# The pan-PPAR agonist lanifibranor improves cardiometabolic health in patients with metabolic dysfunction-associated steatohepatitis

Michael P. Cooreman ®[1,2] ✉, Javed Butler ®[3], Robert P. Giugliano[4], Faiez Zannad ®[5], Lucile Dzen[1,2], Philippe Huot-Marchand[1,2], Martine Baudin[1,2], Daniel R. Beard[6], Jean-Louis Junien[1,2], Pierre Broqua[1,2], Manal F. Abdelmalek[7,9] & Sven M. Francque ®[8,9]

Lanifibranor, a pan-PPAR agonist, improves liver histology in patients with metabolic dysfunction-associated steatohepatitis (MASH), who have poor cardiometabolic health (CMH) and cardiovascular events as major mortality cause. NATIVE trial secondary and exploratory outcomes (ClinicalTrials.gov NCT03008070) were analyzed for the effect of lanifibranor on IR, lipid and glucose metabolism, systemic inflammation, blood pressure (BP), hepatic steatosis (imaging and histological grading) for all patients of the original analysis. With lanifibranor, triglycerides, HDL-C, apolipoproteins, insulin, HOMA-IR, HbA1c, fasting glucose (FG), hs-CRP, ferritin, diastolic BP and steatosis improved significantly, independent of diabetes status: most patients with prediabetes returned to normal FG levels. Significant adiponectin increases correlated with hepatic and CMH marker improvement; patients had an average weight gain of 2.5 kg, with 49% gaining ≥2.5% weight. Therapeutic benefits were similar regardless of weight change. Here, we show that effects of lanifibranor on liver histology in MASH are accompanied with CMH improvement, indicative of potential cardiovascular clinical benefits.

Metabolic-dysfunction associated steatotic liver disease (MASLD, also known as non-alcoholic fatty liver disease [NAFLD]), the accumulation of lipid-laden vacuoles in hepatocellular cytoplasm, is referred to as the hepatic manifestation of metabolic syndrome (MetS), a cluster of conditions encompassing insulin resistance (IR), dyslipidemia, altered glycemic control, (visceral) adiposity and arterial hypertension[1]. Metabolic-dysfunction associated steatohepatitis (MASH, also known as NASH), its progressive form, is characterized by histological steatohepatitis, i.e., lobular inflammation and hepatocyte injury[2]. MASLD global prevalence is 30% and increasing while MASH prevalence is estimated at 5.27%[3]. With no fully approved pharmacological therapy, MASH remains a major medical need and public health burden[4,5].

The pathophysiology of MASH is a complex network of IR, dyslipidemia, altered carbohydrate metabolism, inflammation, hepatocellular injury and, subsequently, fibrogenesis[2,6]. Increased influx of fatty

[1]Research and Development, Inventiva, New York, NY, USA. [2]Research and Development, Inventiva, Daix, France. [3]Baylor Scott and White Research Institute, Dallas, TX, USA. [4]Brigham and Women's Hospital, Massachusetts General Hospital, Harvard Medical School, Boston, MA, USA. [5]Centre d'Investigations Cliniques Plurithématique 1433, Université de Lorraine, Nancy, France. [6]Translational Medicine Academy, Basel, Switzerland. [7]Division of Gastroenterology and Hepatology, Mayo Clinic, Rochester, MN, USA. [8]Department of Gastroenterology and Hepatology, Antwerp University Hospital and University of Antwerp, Antwerp, Belgium. [9]These authors jointly supervised this work: Manal F. Abdelmalek, Sven M. Francque. ✉e-mail: Michael.Cooreman@inventivapharma.com

acids (FA) into hepatocytes from IR-related adipose tissue dysfunction represents a major upstream mechanism of steatohepatitis[2,7]. Hepatocellular mitochondrial dysfunction, i.e., compromised β-oxidation and electron transport chain activity, induce lipotoxicity by generating reactive lipid intermediaries such as diacylglycerol and ceramides, that reinforce IR and damage cellular structures including endoplasmic reticulum and mitochondria. Cellular injury and resulting inflammation, described as "disease activity", drive fibrogenesis, a downstream phenomenon in the disease biology[8,9].

Without management, liver fibrosis can progress to cirrhosis and ultimately end-stage liver disease[10]. The histological fibrosis stage, expressed categorically on a 0–4 scale, predicts liver-related mortality. Patients who progressed to cirrhosis, histological F4 per NASH Clinical Research Network (NASH-CRN) staging[11], have significantly increased mortality from complications of decompensated cirrhosis, versus patients with pre-cirrhotic fibrosis[12–14].

The long natural history of MASLD/MASH is typically accompanied by impaired cardiometabolic health (CMH) with increased risk for cardiovascular (CV) disease (CVD) and type-2 diabetes (T2D), including IR, atherogenic lipid profile, poor glycemic control, systemic inflammation and arterial hypertension[15]. A large body of evidence shows that patients with MASLD are at increased risk for subclinical CVD as well as for clinical CV events and death, to a considerable extent explained by the same underlying disease biology that drives MASLD progression[16]. Steatosis in itself adds to the CVD risk and is pathogenetically linked to atherosclerosis as a source of atherogenic lipids[17–19]. The risk increases further in patients with MASH and MASH with fibrosis compared to steatosis only[20,21]. Other mechanisms add to the link between MASH and atherosclerosis, including prothrombotic and inflammatory mediators and angiogenetic factors[22].

CV events are the most frequent cause of mortality in MASH and begin to occur before advanced liver disease manifests[17,23–27]. Therefore, an investigational compound for MASH ideally shows efficacy on both hepatic and CV health and its effects on CMH markers may inform on its potential to improve CV outcomes.

Peroxisome proliferator-activated receptors (PPAR) are nuclear transcription factors regulating lipid and carbohydrate metabolism, inflammation and fibrosis in liver, adipose tissue, skeletal muscle, and other tissues through programmatic regulation of gene expression. PPARs exist in three isoforms, α, β/δ, and γ. PPARs have distinct yet overlapping tissue distribution and functions, including FA transport and mitochondrial β-oxidation; apolipoprotein (APO) production (mainly PPARα); immune and skeletal muscle homeostasis (mainly PPARβ/δ); insulin sensitivity (IS) in adipose tissue, liver and skeletal muscle; and inhibiting fibrogenesis (mainly PPARγ)[2]. PPAR transcription factors are thus master regulators of the spectrum of MASH disease biology, from IR and metabolic-immune pathways to fibrosis[28].

Lanifibranor is a pan-PPAR agonist with balanced α, β/δ, and γ activity[29]. It has a distinct chemical structure from other PPAR agonists and a unique pattern of co-activator or -repressor recruitment upon ligand binding[30]. These characteristics comprehensively address MASH biology and make lanifibranor a promising investigational compound for this indication. Efficacy and safety of lanifibranor in patients with pre-cirrhotic MASH was demonstrated in the phase 2b study NATIVE. Results focused on hepatic endpoints, including both MASH resolution and fibrosis improvement, as well as safety, were recently published[31]. Here we show that lanifibranor improves CMH in patients with MASH.

## Results

### Baseline characteristics

Of 247 patients enrolled, 228 completed the study; mean age was 54 years, 58% were female, 94% were Caucasian, mean body mass index (BMI) was 33 kg/m²; 42% ($n = 103$) had T2D. Of these, 83% received metformin (alone or in combination anti-diabetic treatment), 22%, sulphonylurea; 11%, dipeptidyl peptidase-4 inhibitor; 7% sodium-glucose cotransporter-2 inhibitor medication; 13% had no anti-diabetic treatment.

Among 144 patients without T2D, 47 (33%) had prediabetes. Overall, 20% of patients took statins (22%, 22%, 16% with lanifibranor 1200 mg, 800 mg, and placebo respectively). Baseline fasting low density lipoprotein cholesterol (LDL-C) levels were lower in patients on statins (mean close to 2.3 mmol/L) than without (3.0 mmol/L). The cardiometabolic profile and characteristic of MetS were similar in all treatment groups (Supplementary Table 2).

### Insulin resistance and glycemic control

Fasting insulin levels (FIL) significantly decreased with lanifibranor in the overall population; adjusted mean difference versus placebo [AMD] at end of treatment [EOT] was −79 pmol/L (95% confidence interval [CI]: −112 to −47, $p < 0.001$) with lanifibranor 1200 mg; −83 pmol/L (95%CI: −115 to −51, $p < 0.001$) with 800 mg. In patients with T2D, AMD was −68 pmol/L (95%CI: −125 to −12, $p = 0.018$) with 1200 mg; −106 pmol/L (95%CI: −163 to −49, $p < 0.001$) with 800 mg; In non-T2D patients, AMD was −86 pmol/L (95%CI: −122 to −50, $p < 0.001$) with 1200 mg; −70 pmol/L (95%CI: −105 to −34, $p < 0.001$) with 800 mg. Similarly, lanifibranor significantly reduced homeostatic model assessment of IR (HOMA-IR) overall (AMD of −4.1 [95% CI: −5.8 to −2.4] with 1200 mg; −4.0 [95%CI: −5.7; −2.3] with 800 mg, both $p < 0.001$), with improvements in the T2D population (AMD of −4.5 [95%CI: −7.8 to −1.2], $p = 0.008$ with 1200 mg; −6.3 [95%CI: −9.7 to −2.8], $p < 0.001$ with 800 mg) and the non-T2D population (Table 1).

Fasting glucose (FG) levels significantly decreased with lanifibranor in the overall population (Fig. 1E). In T2D, the AMD at EOT was −1.2 mmol/L (95%CI: −1.9 to −0.5, $p < 0.001$) with 1200 mg and −1.7 mmol/L (95%CI: −2.4 to −1.0, $p < 0.001$) with 800 mg. Overall, AMD was −0.8 mmol/L (95%CI: −1.2 to −0.5, $p < 0.001$) with 1200 mg and −1.0 mmol/L (95%CI: −1.4 to −0.7, $p < 0.001$) with 800 mg. With T2D, AMD in hemoglobin $A_{1c}$ (HbA1c) at EOT was −0.7% (95%CI: −1.0 to −0.4, $p < 0.001$) in both active arms. Overall, HbA1c was −0.5% (95%CI: −0.6 to −0.3, $p < 0.001$) with both lanifibranor 1200 mg and 800 mg (Table 1, Fig. 1F).

### Glycemic control in patients with prediabetes

With prediabetes, FG decreased at EOT to <5.6 mmol/L in 71% (95%CI: 48 to 95, $p = 0.009$) of patients on lanifibranor 1200 mg, 67% (95%CI: 45 to 88, $p = 0.013$) on 800 mg versus 11% (95%CI: 0 to 32) on placebo, indicating return to normoglycemia in the majority of lanifibranor-treated patients. At EOT, 86% (95%CI: 67 to 100, $p = 0.05$) of patients on lanifibranor 1200 mg and 78% (95%CI: 59 to 97, $p = 0.08$) on 800 mg returned to normal FIL (≤173 pmol/L) versus 37% (95%CI: 4 to 71) on placebo. HOMA-IR normalized (≤3) in 21% (95%CI: 0 to 43) and 23% (95%CI: 3 to 44) with lanifibranor 1200 mg and 800 mg, respectively, versus none on placebo ($n = 8$) (Supplementary Table 3).

Conversely, among 83 patients with normoglycemia at baseline, 0% with lanifibranor 1200 mg and 800 mg progressed to prediabetes at EOT, versus 26% (95%CI: 10 to 41, both $p < 0.01$) in the placebo arm (Supplementary Table 3).

### Lipid metabolism

Fasting total serum triglycerides (TG) levels decreased significantly with lanifibranor compared to placebo, with an AMD at EOT of −0.5 mmol/L (95%CI: −0.8 to −0.3, equivalent to −44.3 mg/dL, $p < 0.001$) with both lanifibranor 1200 mg and 800 mg groups (Table 1). Among the 109 patients with high baseline TG values (>1.7 mmol/L, 150.5 mg/dL), 60% and 70% on lanifibranor 1200 mg and 800 mg, respectively, improved to low-risk TG levels of ≤1.7 mmol/L at EOT, versus 27% on placebo (Supplementary Fig. 1).

**Table 1 | Lanifibranor treatment effect versus placebo on CMH markers (n = 247 patients)**

| Category Parameters (unit) Diabetic status | Adjusted mean difference versus placebo at EOT (SE), 95%CI, two-sided MMRM p value | |
| --- | --- | --- |
| | Lanifibranor 800 mg $N_{Overall}$ = 83/$N_{T2D}$ = 33 / $N_{non-T2D}$ = 50 | Lanifibranor 1200 mg $N_{Overall}$ = 83/$N_{T2D}$ = 35/ $N_{non-T2D}$ = 48 |
| Insulin resistance | | |
| FIL (pmol/L) | −82.96 (16.38), [−115.27; −50.65], <0.001 | −79.21 (16.53), [−111.82; −46.6], <0.001 |
| T2D patients | −105.76 (28.72), [−162.92; −48.60], <0.001 | −68.43 (28.23), [−124.6; −12.25], 0.018 |
| non-T2D patients | −69.61 (17.73), [−104.75; −34.46], <0.001 | −85.88 (18.24), [−122.04; −49.72], <0.001 |
| HOMA–IR[a] | −3.98 (0.86), [−5.68; −2.29], <0.001 | −4.12 (0.86), [−5.82; −2.41], <0.001 |
| T2D patients | −6.25 (1.71), [−9.68; −2.81], <0.001 | −4.54 (1.65), [−7.83; −1.24], 0.008 |
| non−T2D patients | −2.77 (0.76), [−4.28; −1.26], <0.001 | −3.45 (0.79), [−5.01; −1.89], <0.001 |
| Glycemic control | | |
| FG (mmol/L) | −1.02 (0.16), [−1.35; −0.70], <0.001 | −0.84 (0.16), [−1.16; −0.52], <0.001 |
| T2D patients | −1.67 (0.34), [−2.35; −1.00], <0.001 | −1.21 (0.34), [−1.88; −0.54], <0.001 |
| HbA1c (%) | −0.45 (0.07), [−0.59; −0.32], <0.001 | −0.49 (0.07), [−0.62; −0.35], <0.001 |
| T2D patients | −0.69 (0.13), [−0.95; −0.42], <0.001 | −0.73 (0.13), [−0.99; −0.47], <0.001 |
| Lipid metabolism and apolipoprotein levels | | |
| TG (mmol/L) | −0.55 (0.13), [−0.79; −0.3], <0.001 | −0.50 (0.12), [−0.74; −0.25], <0.001 |
| HDL-C (mmol/L) | 0.16 (0.03), [0.09; 0.22], <0.001 | 0.10 (0.03), [0.03; 0.16], 0.003 |
| LDL-C (mmol/L) | 0.02 (0.1), [−0.18; 0.21], 0.875 | 0.01 (0.1), [−0.18; 0.2], 0.897 |
| APO-A1 (mg/dL) | −0.89 (4.46), [−9.76; 7.98], 0.842 | −6.78 (4.24), [−15.2; 1.65], 0.113 |
| APO-B (mg/dL) | −9.66 (2.87), [−15.32; −3.99], <0.001 | −9.76 (2.85), [−15.38; −4.14], <0.001 |
| APO-B/APO–A1 | −0.08 (0.03), [−0.13; −0.03], 0.002 | −0.06 (0.02), [−0.11; −0.01], 0.013 |
| APO-C3 (ug/mL) | −18.38 (5.57), [−29.35; −7.41], 0.001 | −20.29 (5.5), [−31.14; −9.44], <0.001 |
| Systemic inflammation | | |
| hs–CRP (mg/L) | −2.16 (0.66), [−3.46; −0.86], 0.001 | −1.48 (0.66), [−2.77; −0.18], 0.026 |
| Ferritin (µg/L) | −84.24 (20.76), [−125.16; −43.33], <0.001 | −71.81 (20.70), [−112.61; −31.02], <0.001 |
| Liver tests | | |
| ALT (U/L) | −24.69 (5.46), [−35.45; −13.93], <0.001 | −23.14 (5.44), [−33.86; −12.42], <0.001 |
| AST (U/L) | −15.03 (4.54), [−23.99; −6.08], 0.001 | −11.96 (4.51), [−20.85; −3.07], 0.009 |
| GGT (U/L) | −47.79 (7.96), [−63.51; −32.07], <0.001 | −32.28 (7.92), [−47.93; −16.63], <0.001 |
| Blood Pressure | | |
| Diastolic BP (mmHg) | −3.89 (1.54), [−6.93; −0.85], 0.012 | −2.45 (1.53), [−5.47; 0.56], 0.110 |
| Systolic BP (mmHg) | −2.47 (2.16), [−6.72; 1.79], 0.255 | −0.38 (2.14), [−4.60; 3.84], 0.859 |
| NT–proBNP (pmol/L) | 3.98 (1.62), [0.78; 7.17], 0.015 | 8.42 (1.58), [5.31; 11.54], <0.001 |
| Steatosis | | |
| CAP™ (dB/m) | −16.06 (8.80), [−33.46; 1.33], 0.070 | −23.25 (9.17), [−41.37; −5.12], 0.012 |

*APO* apolipoprotein, *ALT* alanine aminotransferase, *AST* aspartate aminotransferase, *BP* blood pressure, *CAP™* controlled attenuation parameter, *CI* confidence interval, *CMH* cardiometabolic health, *EOT* end of treatment, *FG* fasting glucose, *FIL* fasting insulin levels, *GGT* gamma–glutamyl transferase, *HbA1c* hemoglobin A1c, *HDL* high density lipoprotein, *HOMA-IR* homeostatic model assessment of insulin resistance, *hs-CRP* high-sensitivity C-reactive protein, *LDL* low density lipoprotein, *MMRM* mixed model for repeated measures, *NT-proBNP* N-terminal pro b-type natriuretic peptide, *SE* standard error, *T2D* type-2 diabetes, *TG* Triglycerides.
[a]Patients treated with sulphonylureas were removed from HOMA–IR related analyses. Resulting from MMRM models using change from baseline as endpoint, the time, treatment, the diabetic status, the interaction (treatment*time) and the baseline value as fixed effects, a time repeated measure within each subject and an unstructured variance covariance matrix; no adjustment for multiple comparisons was performed. Similar results were obtained when considering concomitant use of metformin and statins as covariates in the statistical model [results not shown].

Fasting high density lipoprotein cholesterol (HDL-C) increased significantly at treatment week 4 (TW4), maintained until EOT: patients receiving lanifibranor 1200 mg and 800 mg had an AMD at EOT of 0.10 mmol/L (95%CI: 0.03 to 0.16, p = 0.003, equivalent to 3.9 mg/dL) and 0.16 mmol/L (95%CI: 0.09 to 0.22, p < 0.001, equivalent to 6.2 mg/dL), respectively (Table 1). TG and HDL-C changes with lanifibranor were consistent over time (Fig. 1A, B). No significant LDL-C change was observed with either lanifibranor dose versus placebo (Table 1, Fig. 1C).

Lanifibranor significantly reduced APO-B with an AMD at EOT of −10 mg/dL (95%CI: −15 to −4, p < 0.001) with both doses (Table 1, Fig. 1D), as well as the APO-B/APO-A1 ratio with an AMD at EOT of −0.06 (95%CI: −0.11 to −0.01, p = 0.013) with 1200 mg and −0.08 (95%CI: −0.13 to −0.03, p = 0.002) with 800 mg (Table 1). APO-A1 levels did not change significantly at EOT.

**Systemic inflammation**
Baseline high-sensitivity C-reactive protein (hs-CRP) levels were elevated in most patients: mean 5.4, 5.1 and 4.0 mg/dL in lanifibranor 1200 mg, 800 mg, and placebo arms, respectively (Supplementary Table 2). Values decreased at EOT in both lanifibranor arms but remained stable under placebo (Fig. 1G). With lanifibranor 1200 mg and 800 mg respectively, AMD at EOT was −1.5 mg/L (95%CI: −2.8 to −0.2, p = 0.026) and −2.2 mg/L (95%CI: −3.5 to −0.9, p = 0.001); (Table 1); 38% (95%CI: 22 to 53) and 44% (95%CI: 28 to 61) of patients at high CV risk improved to intermediate or low risk and 44% (95%CI: 25 to 63) and 35% (95%CI: 16 to 53) of those at intermediate risk improved to low risk, respectively (Supplementary Table 4). Only 26% (95%CI: 12 to 40) of patients on placebo at high risk improved to intermediate risk and 13% (95%CI: 0 to 27) improved from intermediate to low risk.

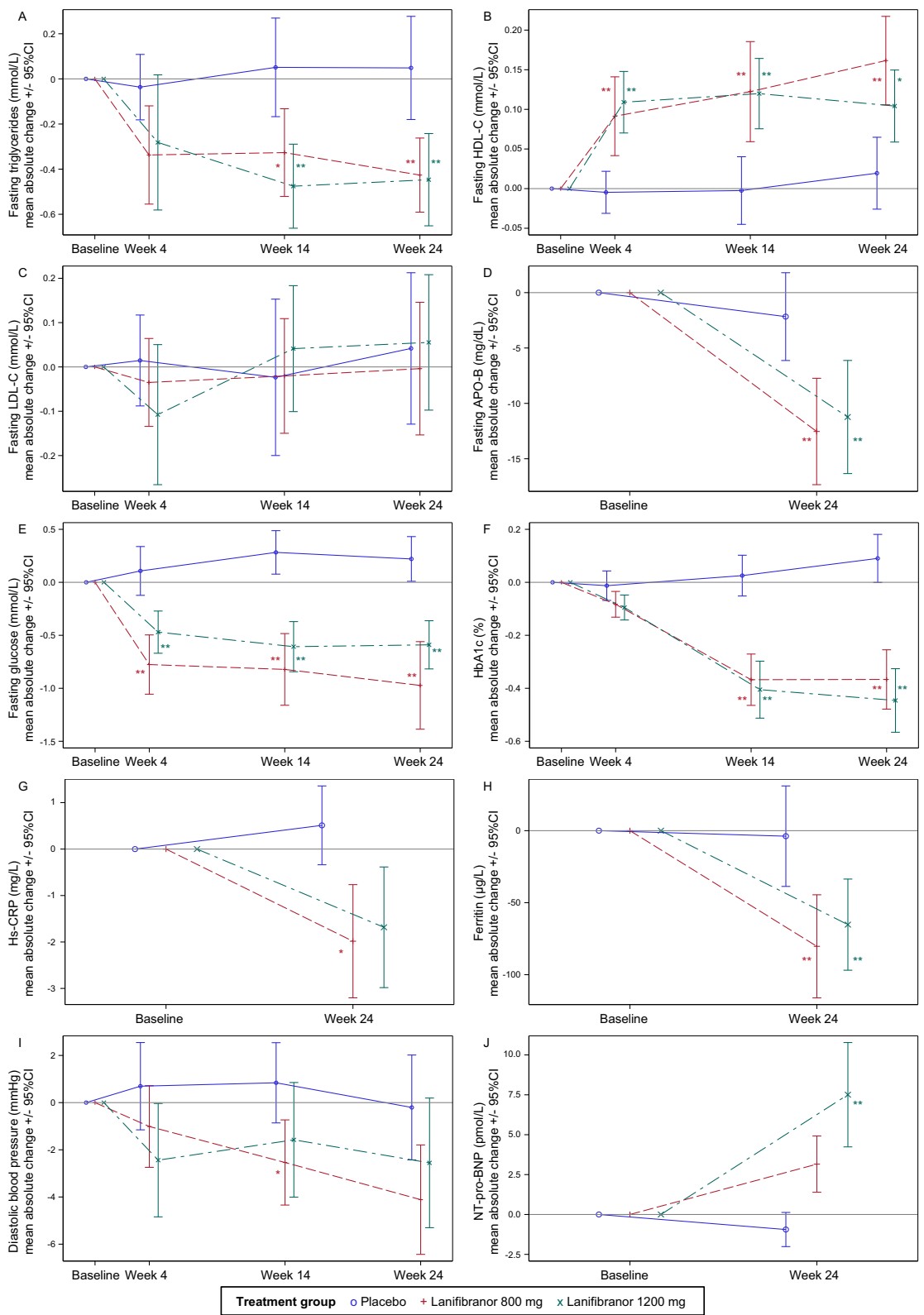

**Treatment group**  o Placebo   + Lanifibranor 800 mg   x Lanifibranor 1200 mg

Mean baseline ferritin levels were comparably elevated: 237.5 (standard deviation [SD]: 238.2) µg/L, 230.9 (SD: 229.0) µg/L and 254.6 (SD: 233.3) µg/L with lanifibranor 800 mg, 1200 mg and placebo, respectively (Supplementary Table 2). At EOT, levels decreased significantly with lanifibranor (AMD at EOT: −84.2 µg/L [95%CI: −125.2 to −43.3], −71.8 µg/L [95%CI: −112.6 to −31.0] for lanifibranor 800 mg and

1200 mg, respectively, both $p < 0.001$), but were unchanged with placebo (Table 1, Fig. 1H).

**Blood pressure (BP)**

From TW4 and maintained during treatment, diastolic BP (DBP) decreased with both lanifibranor doses (AMD at EOT was −2.5 mmHg

**Fig. 1 | Time courses for CMH and hepatic markers absolute change from baseline, by to treatment groups. A** Triglycerides (*n* = 242 patients at W4, 231 at W14 and 217 at EOT), **B** HDL-C (*n* = 243 patients at W4, 232 at W14 and 218 at EOT), **C** LDL-C (*n* = 239 patients at W4, 227 at W14 and 213 at EOT), **D** APO-B (*n* = 208 patients), **E** fasting glucose (*n* = 242 patients at W4, 232 at W14 and 216 at EOT), **F** HbA1c (*n* = 243 patients at W4, 233 at W14 and 218 at EOT), **G** hs-CRP (*n* = 218 patients), **H** ferritin (*n* = 218 patients), **I** diastolic blood pressure (*n* = 243 patients at W4, 233 at W14 and 218 at EOT) and **J** NT-proBNP (*n* = 212 patients). Data are plotted as mean ± 95%CI. Treatment groups are indicated as follows: placebo, blue circle and solid line; lanifibranor 800 mg, red plus and dash line; lanifibranor 1200 mg,

green cross and mixed line. \**p* < 0.01, \*\**p* < 0.001, from two-sided Mixed Model for Repeated Measures using change from baseline as endpoint, the time, treatment, the diabetic status, the interaction (treatment\*time) and the baseline value as fixed effects, a time repeated measure within each subject and an unstructured variance covariance matrix. No adjustment for multiple comparisons was performed. Source data are provided as a Source Data file. APO apolipoprotein, CI confidence interval, EOT end of treatment, HbA1c hemoglobin A1c, HDL high density lipoprotein, hs-CRP high-sensitivity C-reactive protein, LDL low density lipoprotein, *p* = *p* value, NT-proBNP N-terminal pro b-type natriuretic peptide, W week.

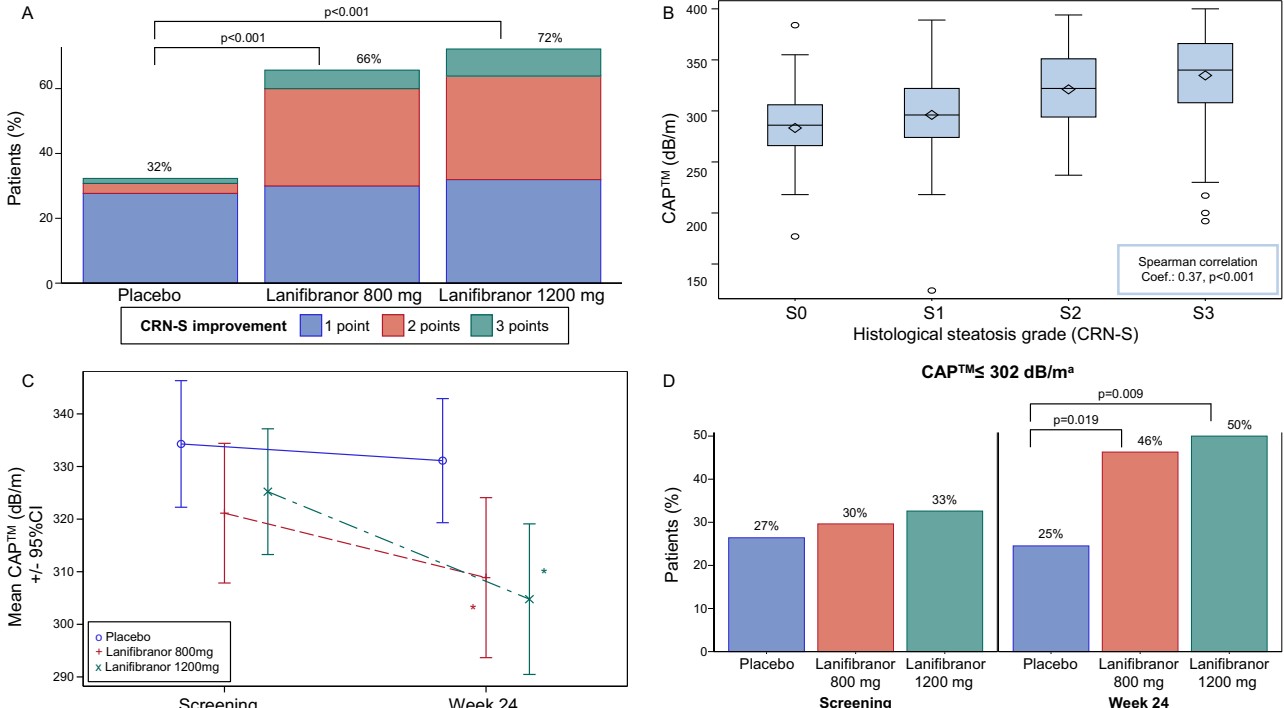

**Fig. 2 | Hepatic steatosis. A** CRN-S improvement, defined as at least one-point improvement of CRN-S at EOT compared to screening (*n* = 207 patients). Point-improvement groups are indicated as follows: 1 point, blue; 2 points, red; 3 points, green. p-values from two-sided Cochran-Mantel-Haenszel test, stratified on the diabetic status at Baseline. **B** Correlation between CAP™ and CRN-S (*n* = 294 pooled pre- and post-treatment biopsies). The box's length represents the IQR, the diamond inside the box represents the mean, the horizontal line inside the box represents the median, the whiskers extend to the minimum and maximum values included in [Q1-1.5\*IQR; Q3 + 1.5IQR], the circles outside the whiskers represent the outliers. *P* value from two-sided Spearman correlation test. **C** CAP™ at screening

and EOT, by treatment group (*n* = 153 patients). Data are plotted as mean ± 95%CI. Treatment groups are indicated as follows: placebo, blue circle and solid line; lanifibranor 800 mg, red plus and dash line; lanifibranor 1200 mg, green cross and mixed line. \**p* < 0.05, from two-sided Wilcoxon-Mann-Whitney test. **D** CAP™ ≤302ª dB/m at screening and EOT (*n* = 153 patients). *P* values from two-sided Chi² test ª Youden optimal cutoff of the CAP™ as a diagnosis marker of steatosis ≥S1 versus S0. No adjustment for multiple comparisons was performed. Source data are provided as a Source Data file. CAP™ controlled attenuation parameter, CI confidence interval, Coef. coefficient, CRN Clinical Research Network, EOT end of treatment, IQR interquartile range.

[95%CI: −5.5 to 0.6, *p* = 0.110] with lanifibranor 1200 mg and −3.9 mmHg [95%CI: −6.9 to −0.9, *p* = 0.012] with 800 mg, [Table 1, Fig. 1I]). Systolic BP did not change significantly.

## Hepatic steatosis
Histological steatosis improved significantly at EOT with lanifibranor versus placebo, with more than 35% of patients improving by at least 2 grades with lanifibranor versus 5% with placebo (Fig. 2A). Controlled attenuation parameter (CAP™) similarly decreased significantly at EOT with lanifibranor (AMD at EOT was −23 dB/m [95%CI: −41 to −5, *p* = 0.012] with 1200 mg and −16 dB/m [95%CI: −33 to 1, *p* = 0.070] with 800 mg [Table 1]) but not with placebo (Fig. 2C). Baseline CAP™ category distribution was comparable between treatment groups; at

EOT, a significantly higher proportion of patients had CAP™ ≤302 dB/m, corresponding to S ≤ S1, with lanifibranor 1200 mg (50%, 95%CI: 36 to 64, *p* = 0.009) and 800 mg (46%, 95%CI: 33 to 60, *p* = 0.019) versus placebo (25%, 95%CI: 13 to 37) (Fig. 2D). There was a significant relationship (Spearman coefficient [Rs]: 0.37, *p* < 0.001) between CAP™ and histological steatosis grade at screening and EOT (Fig. 2B). CAP™ steatosis decrease correlated with TG and HbA1c lowering at EOT (Supplementary Fig. 2C, D).

## Weight change in relation to CMH and hepatic markers
Mean absolute (resp. relative) weight increase at EOT was 2.7 (3.1%) and 2.4 (2.6%) kg for lanifibranor 1200 mg and 800 mg, respectively. Weight change showed considerable interindividual variability: at EOT,

51% of patients in pooled lanifibranor arms had stable weight (≤2.5% increase), 16% had moderate increase (2.5–5%), and 33% had ≥5% increase from baseline; with placebo 84% had stable weight, but 16% had >2.5% weight increase (Supplementary Fig. 3). In pooled lanifibranor arms, all CMH markers, i.e., steatosis, HOMA-IR, FIL, lipids (HDL-C, TG, APO-B, APO-C3, APO-B/APO-A), FG, hs-CRP, ferritin and DBP, as well as liver tests (ALT, AST, gamma-glutamyl transferase [GGT]), improved to the same degree in the three weight-change groups. In contrast, placebo-treated patients with increased weight at EOT had unchanged or worsening values of these markers. (Supplementary Table 5, Fig. 3A, B, C, E, F). N-terminal pro b-type natriuretic peptide (NT-proBNP) increased with both lanifibranor doses; AMD at EOT was 8.4 pmol/L (71 pg/mL) with lanifibranor 1200 mg and 4.0 pmol/L (34 pg/mL) with 800 mg (Table 1, Fig. 1J), this increase being correlated with weight increase (Supplementary Table 4, Fig. 3G).

### Adiponectin, a cardiometabolic and hepatic efficacy marker

Mean baseline adiponectin levels were low at 5.1 µg/mL with high variation (range 0.8–30.8 µg/mL); 60% of patients had low, 32% medium and 8% high values (Supplementary Table 6). Higher increase of adiponectin levels was observed in the lanifibranor 1200 mg (mean fold 4.5) compared to lanifibranor 800 mg (3.8) at EOT, 86% of patients on lanifibranor 800 mg had moderate (58%) or high (29%) increase and 95% of those on 1200 mg had moderate (52%) or high (42%) increase. With placebo only 10% had moderate, and none high increase (Supplementary Table 6). In the pooled lanifibranor arms, adiponectin increased in all three weight-change groups, with a higher increase in the ">2.5% weight increase" groups, versus the "≤2.5% weight stable" group (Fig. 3D).

Adiponectin increase at EOT correlated with improvement in IR (FIL, HOMA-IR), glycemic control (FG, Hb1Ac), lipid metabolism (TG and APO-B decrease, HDL-C increase), hs-CRP, aminotransferases, steatosis markers (GGT, CAP™), with a larger effect size for high than for moderate increase (Supplementary Table 7, Fig. 4A–E). Mean CAP™ values improved from 317 at baseline to 280 dB/m at EOT in patients with >4-fold adiponectin increase, 73% of them to values < 302 dB/m (corresponding to steatosis grade ≤1) (Fig. 4E).

Absolute and categorical (fold change) adiponectin increase also correlated with improvement of liver histological endpoints for disease activity, i.e., CRN-NAFLD activity score (NAS) and individual activity components (ballooning and inflammation), and with improvement in fibrosis stage (38% versus 57% for adiponectin increase ≤4- versus >4-fold, respectively) (Supplementary Table 8, Fig. 5A–D).

### Correlation between improvement of IR and hepatic markers

Baseline HbA1c levels correlated with baseline histological MASH activity and fibrosis (Spearman Rs: 0.15, $p = 0.017$, and 0.15, $p = 0.021$, respectively) (Supplementary Fig. 4). HbA1c improvement correlated with histological response in the lanifibranor arms: HbA1c decrease was larger in histological steatosis "improvers" (−0.46%) versus "non-improvers" (−0.20%, $p = 0.048$); and ballooning "improvers" (−0.45%) versus "non-improvers" (−0.33%, $p = 0.088$) (Supplementary Fig. 5). Similarly, HOMA-IR decrease correlated with improvement of liver tests and normal HOMA-IR at EOT was associated with histological improvement (Supplementary Tables 9, 10).

Under lanifibranor, HbA1c decrease also correlated with improvement of FG (Spearman Rs: 0.55, $p < 0.001$), FIL (Rs: 0.36, $p < 0.001$), HOMA-IR (Rs: 0.51, $p < 0.001$), ALT (Rs: 0.35, $p < 0.001$), AST (Rs: 0.28, $p < 0.001$), GGT (Rs: 0.33, $p < 0.001$), CAP™ (Rs: 0.17, $p = 0.092$), HDL-C (Rs: −0.21, $p = 0.011$), TG (Rs: 0.27, $p < 0.001$), APO-B (Rs: 0.23, $p = 0.008$) and APO-B/APO-A1 (Rs: 0.26, $p = 0.002$) (Supplementary Table 11).

### Correlation between steatosis improvement and CMH markers

Baseline CAP™ steatosis ≤302 and >302 dB/m correlated with HOMA-IR (7.8 and 11.9, $p = 0.003$), HbA1c (5.7 and 6.2%, $p < 0.001$), TG (1.72 and 2.04 mmol/L, $p = 0.012$), DBP, and inversely with HDL-C (1.30 and 1.18 mmol/L, $p = 0.007$) and APO-A1 (151 and 142 mg/dL, $p = 0.027$), respectively; adiponectin did not differ with steatosis severity (Supplementary Table 12). Improvement of HOMA-IR with lanifibranor therapy was independent of steatosis reduction; HbA1c improvement was correlated with improvement of both histological steatosis grade and CAP™ change; TG decrease correlated with change in CAP™ categorized as increase (≥+10%), no change (−10%; +10%) and decrease (< −10%); HDL-C increase also correlated with histological steatosis improvement. Adiponectin levels increased 3.1-, 3.9-, 5.3- and 6.0-fold for no change, 1-, 2- and 3-points reductions of steatosis grade (Spearman Rs: −0.40, $p < 0.001$) (Supplementary Fig. 2A–F).

### Cardiometabolic health markers in obese and non-obese patients

Of 247 patients enrolled, 161 (65%) patients were obese (BMI ≥ 30) and 86 (35%) patients had a BMI < 30 at baseline (Supplementary Table 1). Overall, the effects of lanifibranor therapy versus placebo on markers of CMH and on liver tests were similar for obese and non-obese patients (Supplementary Table 13).

## Discussion

Patients enrolled in the NATIVE trial had, as expected, poor CMH with increased CV risk, namely IR, atherogenic lipid profile, poor glycemic control, systemic inflammation, elevated BP and hepatic steatosis. The data presented here demonstrate that lanifibranor therapy broadly and significantly improves metabolic-immune markers that are associated with the risk for CVD and T2D in patients with MASH, with similar effect sizes for both doses studied.

Elevated TG at baseline corresponded to different degrees of CV risk; after 4 weeks and throughout the treatment period, most patients on lanifibranor reverted to "low-risk range" levels. Similarly, APO-B, APO-C3 and APO-B/APO-A1 were significantly lowered at EOT, and HDL-C increased from week 4 throughout treatment. Improving proatherogenic lipid levels, including TG and APO-B, may reduce CV events[32], and the recognition of APO-C3 as an independent risk factor for atherosclerosis has led to renewed interest in APO-C3 as therapeutic target. The potential aggregate clinical effect from several lipid risk factors moving in the right direction warrants further evaluation of lanifibranor therapy for CV outcomes.

The risk for CVD is further determined by IR, glycemia control, systemic inflammation, BP and steatosis, risk factors that all respond to lanifibranor therapy. Hs-CRP levels decreased significantly with both lanifibranor doses, whereas a mean increase was seen with placebo. Hs-CRP, an acute phase protein produced by hepatocytes, macrophages and vascular smooth muscle cells, is a marker of systemic inflammation associated with formation and rupture of atherosclerotic plaques, promotion of occlusive thrombi, damage of vascular endothelial integrity, activation of the renin-angiotensin system and vascular remodeling[33]. CRP is an important predictor of adverse coronary events independent of dyslipidemia[34–36]. About 50% of patients had high-risk hs-CRP levels (>3 mg/dL); although mean baseline values were higher in the active arms than in the placebo arm, a significantly larger percentage of patients reached hs-CRP levels of a lower risk category with lanifibranor than with placebo. Similarly, elevated ferritin levels are related to inflammation, IR and CVD parallel to the effects of hs-CRP in MetS[37]. Both lanifibranor doses significantly reduce ferritin levels.

Poor glycemic control is also associated with severity of MASH; higher HbA1c values correlate with higher degrees of histological steatosis, ballooning hepatocytes and fibrosis[38–40], a correlation that was also observed in NATIVE. With lanifibranor, HbA1c decrease at EOT

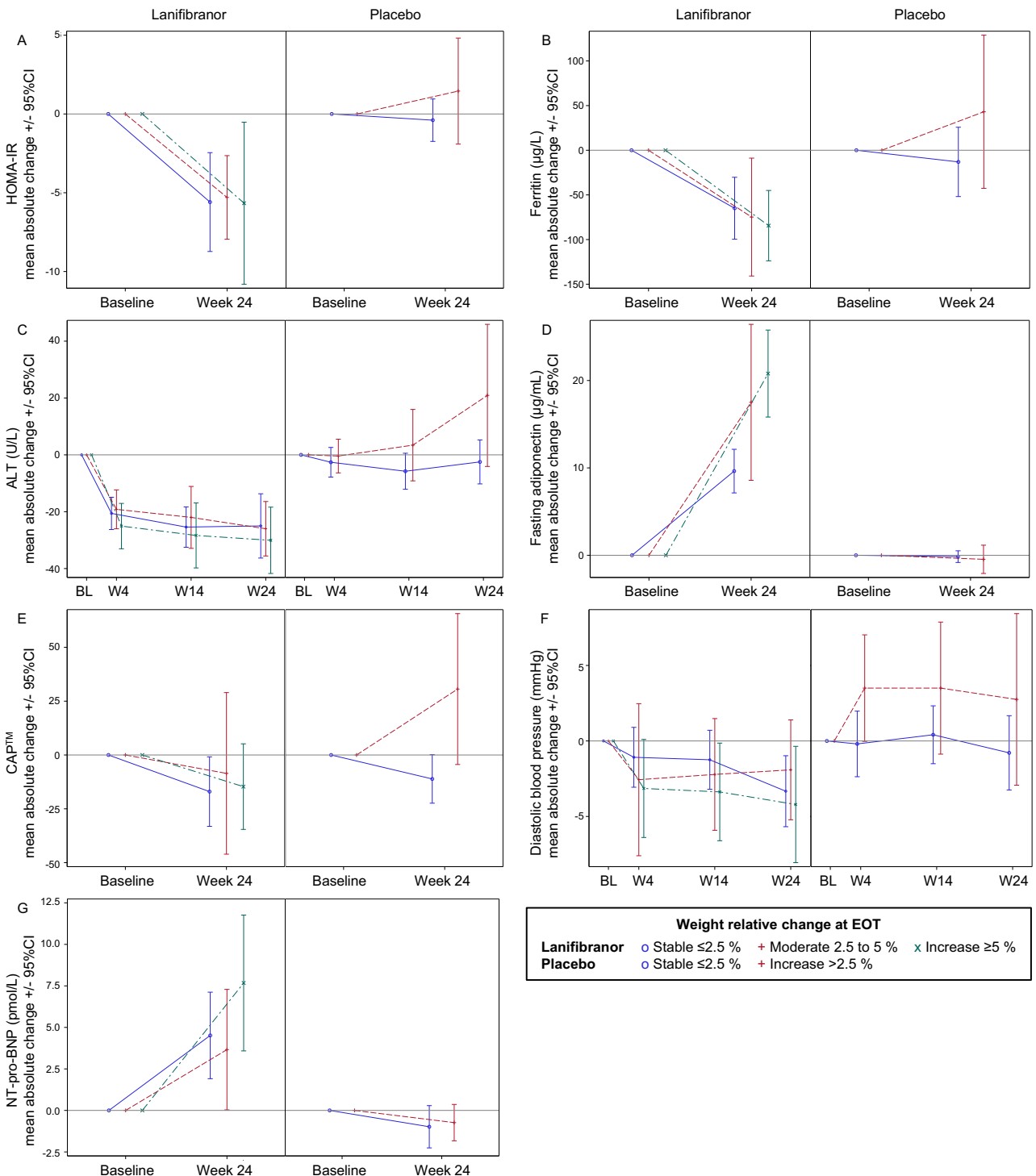

**Fig. 3 | Time courses for CMH and hepatic markers absolute change from baseline according to weight-change groups at EOT, by treatment group.** **A** HOMA-IR (*n* = 169 patients), **B** ferritin (*n* = 217 patients), **C** ALT (*n* = 217 patients at W4 and W14, 216 at EOT), **D** adiponectin (*n* = 217 patients), **E** CAP™ (*n* = 142 patients), **F** diastolic blood pressure (*n* = 217 patients at W4, W14 and EOT) and **G** NT-pro-BNP (n = 210 patients). Data are plotted as mean ± 95%CI. Weight-change groups are indicated as follows: stable (≤2.5%), blue circle and solid line; moderate (within 2.5% and 5%), red plus and dash line; increase (≥5%), green cross and mixed line for lanifibranor-treated patients, and stable (≤2.5%), blue circle and solid line; increase (>2.5%), red plus and dash line for placebo-treated patients. Only two weight relative change groups defined for placebo due to few patients ≥5% (*n* = 4). For HOMA-IR related figures, patients treated with sulphonylureas were removed from the analyses. Source data are provided as a Source Data file. ALT alanine aminotransferase, BL Baseline, CAP™ controlled attenuation parameter, CI confidence interval, EOT end of treatment, HOMA-IR homeostatic model assessment of insulin resistance, NT-proBNP N-terminal pro b-type natriuretic peptide, W week.

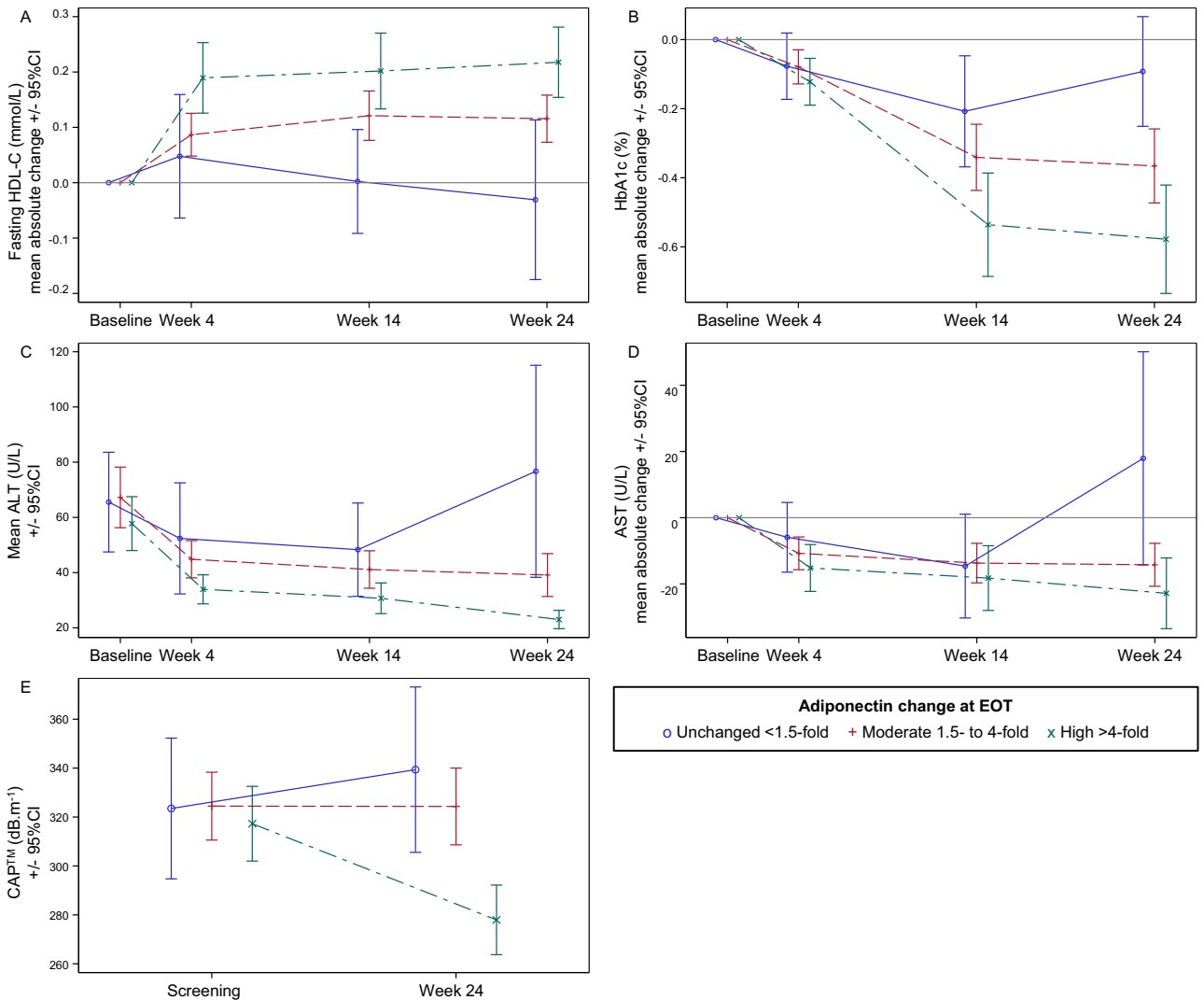

**Fig. 4 | Time courses for CMH and hepatic markers from baseline according to adiponectin-change groups at EOT in lanifibranor treated patients.** Absolute changes in levels of **A** HDL-C, **B** HbA1c, **D** ALT, (n = 139 patients at W4, W14, EOT) and levels of **C** AST (n = 139 patients at baseline, W4, W14, EOT), **E** CAP™ (n = 90 patients at screening and EOT). Data are plotted as mean ± 95%CI. Adiponectin increase groups are indicated as follows: unchanged (<1.5-fold), blue circle and solid line; moderate (within 1.5- and 4-fold), red plus and dash line; high (>4-fold), green cross and mixed line. Source data are provided as a Source Data file. AST aspartate aminotransferase, ALT alanine aminotransferase, CAP™ controlled attenuation parameter, CI confidence interval, EOT end of treatment, HbA1c hemoglobin A1c, HDL high density lipoprotein.

correlated with improvement of metabolic and hepatic (IR, TG, HDL-C, APOs, steatosis, aminotransferases) markers. Among patients without overt T2D, a substantial proportion had prediabetes, which is associated with increased risk for atherosclerotic CVD and later T2D development[41]. Although professional guidelines advocate early management to return FG levels to normal values, effectively preventing T2D development, prediabetes remains underrecognized[42]. Lanifibranor therapy brought FG within the normal range in the majority of patients with prediabetes, thereby reducing the risk for progressive metabolic disease.

Lanifibranor also significantly reduced steatosis, with half of the patients reaching CAP™ values below the threshold for histological grade S1[43], an effect that correlated with lowering of TG and HbA1c.

Weight gain observed with PPAR agonist therapy is well defined as metabolically distinct from life-style-related (poor diet and/or lack of physical activity) weight gain. PPARγ activation reduces visceral and

hepatic fat, while inducing maturation of insulin sensitive subcutaneous fat, as demonstrated with pioglitazone, a PPARγ agonist approved for T2D; in contrast, weight gain from diet failure increased visceral and hepatic fat[44,45]. PPARγ activity thereby induces redistribution from metabolically unhealthy, insulin-resistant abdominal fat to insulin-sensitive, subcutaneous adipose tissue associated with improved metabolic health[46–48]. Lanifibranor data are in alignment with these observations. About half of the patients had weight gain ≥2.5% during lanifibranor therapy, with mean increase of 2.4 and 2.7 kg for low and high dose, respectively. Steatosis, IS and the panel of CMH markers improved in patients on lanifibranor whether or not they had a change in weight, and independent of the degree of weight change. Patients on placebo who gained weight showed either no effect or worsening of CMH markers.

While PPARγ-associated weight gain is thus indicative of maturation of adipose tissue, some contributing effect on fluid retention is suggested by the treatment effect on NT-proBNP. Past pure PPARγ

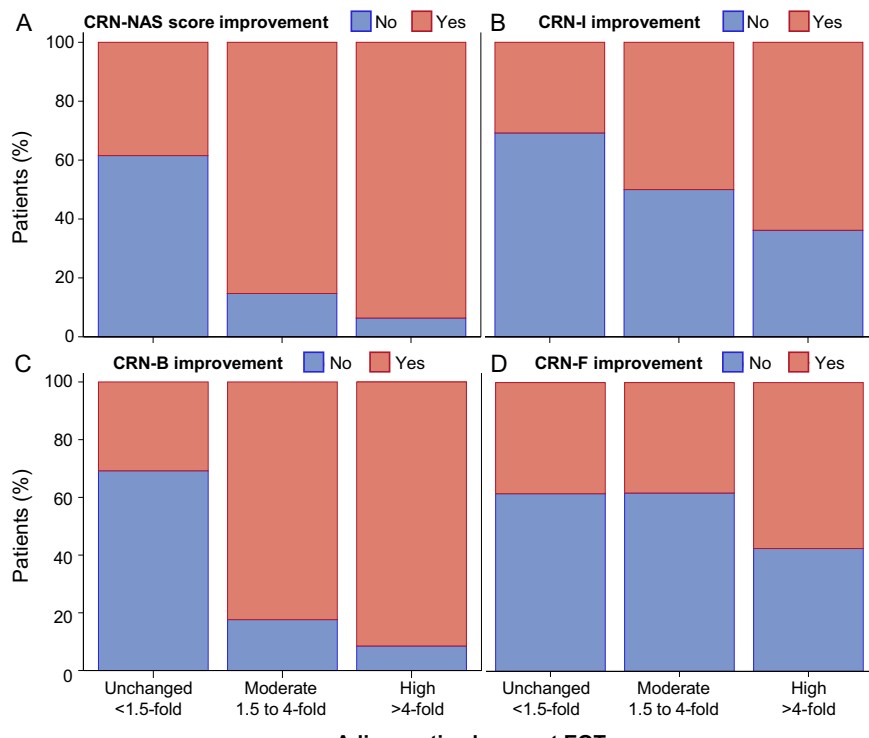

**Fig. 5 | Histological improvement according to adiponectin-change groups at EOT, in lanifibranor-treated patients. A** CRN-NAS score, **B** CRN-I, **C** CRN-B and **D** CRN-F (*n* = 128 patients). Histological improvement is indicated as follows: improvement of at least one point at EOT compared to screening according to NASH-CRN grading and staging, red; and no improvement otherwise; blue.

Adiponectin increase groups are defined as unchanged (<1.5-fold), moderate (within 1.5- and 4-fold) and high (>4-fold). Source data are provided as a Source Data file. B ballooning, CRN Clinical Research Network, EOT end of treatment, F fibrosis, I inflammation, NAS NAFLD activity score.

agonists such as the thiazolidinediones pioglitazone and rosiglitazone are effective for glycemic control and improving IR, but also have been associated with an increased risk of edema and heart failure[49]. There was no clinical evidence for congestion in patients on lanifibranor in NATIVE.

The efficacy of lanifibranor on CMH markers parallels a robust increase in adiponectin levels. Adiponectin is a pleiotropic adipokine and PPARγ downstream mediator, while PPARα activity may affect adiponectin signaling through receptor upregulation[50]. Adiponectin improves IR, lipid metabolism, inflammation and fibrosis; it induces maturation of insulin-sensitive, mostly subcutaneous adipose tissue resulting in less toxic lipid intermediaries[51,52]. Low adiponectin levels are indicative of adipose tissue dysfunction[53], are associated with CVD and have also been reported in MASH[44,54,55], which illustrates the role of adipose tissue dysfunction in both MASH and CVD biology. Baseline levels were indeed low in the patient population of NATIVE. Lanifibranor treatment significantly increased adiponectin levels in >90% of patients, which is expected to lead to maturation of insulin-sensitive adipose tissue, and related weight change[53,56,57]. The degree of adiponectin increase indeed correlated with weight change, as well as with improvement of CMH markers, including IR, lipids, hs-CRP and hepatic steatosis. No adiponectin increase was seen with placebo, rather a small decrease in patients who gained weight on placebo, further pointing to the beneficial role of adiponectin in the opposite metabolic effects correlated with weight change in lanifibranor versus placebo arms. This is supported by the observation that steatosis decreased most in patients with the highest, >4-fold increase of adiponectin levels. The data support a link between adiponectin increase with lanifibranor and a shift toward a metabolically healthier, insulin-sensitive profile.

MASLD and CVD are pathogenetically linked and not just comorbidities resulting from shared risk factors[58]. The association between MASLD and atherosclerosis is stronger than can be explained by common risk factors alone[59]. With CVD being the main cause of mortality, the risk for non-fatal CV events is even stronger in MASLD compared to controls, with a hazard ratio of 3.71[60]. Steatosis by itself increases the risk of atherogenic dyslipidemia and is correlated with the severity of coronary heart disease[61]. In one cohort study, an association between MASLD severity on ultrasound and carotid atherosclerosis as well as between MASLD regression during follow-up and reduced risk for subclinical carotid atherosclerosis was found[62]. Progression to steatohepatitis increases the risk for CVD further[63]. The pathogenetic link between MASLD and CVD results from production and secretion of toxic lipid intermediates and atherogenic lipids in e.g., very low-density lipoproteins by the liver, and from systemic inflammation that accompanies progressive MASH[64].

The broad therapeutic effect of lanifibranor in patients with MASH can in the first instance be explained through its balanced pan-PPAR agonist activity. PPAR isoforms α, β/δ and γ have distinct and overlapping ligand binding, tissue expression pattern and biological effects[2]. Given their crucial biological roles in metabolic-immune diseases, PPARs have been targets of high interest for pharmacological drug development[28]. Among other effects, PPARα agonists reduce TG levels by increasing mitochondrial FA β-oxidation; PPARβ/δ ligands increase HDL-C levels, have anti-inflammatory effects and improve glucose utilization of skeletal muscle, while PPARγ activity exerts antifibrotic effects on activated hepatic stellate cells and improves IS in liver, adipose tissue and skeletal muscle. Together, the PPAR isoforms regulate pathways that encompasses the entire disease biology of MASLD/MASH. This provides the rationale for a PPAR agonist that, in

contrast to selective PPAR agonists such as fibrates (PPARα) and thiazolidinediones (PPARγ), has balanced activity on all PPAR isoforms and is thus expected to have direct therapeutic effects on all major nodes of MASH disease biology, from upstream IR, dysregulated lipid and glucose metabolism, inflammation to hepatic fibrogenesis, and thereby to have therapeutic potential for both hepatic and cardio-metabolic manifestations of MASH. The clinical metabolic-immune improvements with lanifibranor indeed substantiate the pan-PPAR activating mechanism across all three subtypes, exemplified by the effects on TG, HDL-C and IR reflecting mainly PPARα, β/δ and γ mediated mechanisms, respectively[2].

Given the pathway interconnectedness of MASLD and CVD, improvement in hepatic health may have an effect on CMH and vice versa, in addition to direct pan-PPAR mediated pharmacological effects. Although this remains speculative and warrants further research, bidirectional interactions between course and severity of T2D and MASH are well documented[65]. The correlation analyses between improvement of CMH markers such as IR and hs-CRP and markers of hepatic health including steatosis, and histological MASH activity observed in this study point toward multidirectional interactions in disease biology that may add to the benefits of pan-PPAR agonist therapy.

The study has strengths and limitations. Strengths include the randomized double-blind design and the well-characterized, per protocol collected data with two active doses versus placebo in a study of sufficient sample size to show changes in biomarkers and clinical measurements across the different dimensions of CMH. The therapeutic effect sizes are similar for both doses, while the higher dose has a superior effect size on histological liver fibrosis; it will thus be of interest to evaluate the effects on liver histology of both doses after a longer treatment period. Limitations may include the fact that patients who meet study entry criteria may not represent all patients seen in varying practice environments; analyses were post-hoc of available data; other potential biomarkers such as NT-proBNP, proteomics, and hepatokines that affect the intricate network of interactions between the steatotic liver and other endocrine organs may be informative in future studies[66]. Lastly, a longer duration of therapy would be required to relate the effects on CMH markers to potential outcome effects; thus, this study represents a lead-in for combined investigations of hepatic and CV health in MASH.

In conclusion, MASH is part of a systemic metabolic-immune disorder with complex multidirectional interactions between affected tissues. Both progressive liver disease and CVD develop in this context and need to be addressed by pharmacological treatment claiming efficacy. The pan-PPAR agonist lanifibranor significantly improves CMH in correlation with improvements in liver health and indicators of improved adipose tissue function; the improvement in cardiometabolic and hepatic markers was independent of changes in weight, and was correlated with a response of adiponectin, a mediator of improved IS. The NATIVE data demonstrate that lanifibranor is a promising investigational therapy that addresses the broad disease biology of MASH and related CVD, corresponding to its pan-PPAR agonist mechanism of action. Future studies of longer duration are warranted to investigate the effects of lanifibranor on CV outcomes.

## Methods

### Trial oversight
This clinical trial has been registered at ClinicalTrials.gov, number NCT03008070 (first posted January 2, 2017). This study and all amendments have been approved by independent ethics committees and the appropriate authorities (Supplementary Table 1) in 16 countries (France, United States of America, Australia, Belgium, Bulgaria, Germany, Canada, Italy, Spain, Poland, the United Kingdom, Czech Republic, Switzerland, Slovenia, Austria and Mauritius), where at least one patient underwent randomization. The trial was conducted in accordance with the principles of the Declaration of Helsinki, the International Council for Harmonisation Good Clinical Practice guidelines, and all relevant regulations. Prior to the trial entry, written informed consent was obtained from all participants.

### Study population
The NATIVE trial was conducted from 07-Feb-2017 (first patient first visit) to 16-Mar-2020 (last patient last visit), and enrolled 247 patients ≥18 years old with MASH confirmed by a centrally read biopsy performed at screening or in the preceding six months and a histological Steatosis Activity Fibrosis (SAF) score for disease activity (inflammation and 'ballooning') ≥3[67]. Patients with cirrhosis, HbA1c > 8.5% (69 mmol/mol), recent change in anti-diabetic medication, type-1 diabetes or T2D on insulin therapy and other causes of chronic liver disease, including significant daily alcohol consumption, were excluded[31]. A total of 247 patients were randomized 1:1:1 to receive lanifibranor 1200 mg, 800 mg, or placebo, once daily for 24 weeks.

### Cardiometabolic assessments
CMH markers assessed included: (a) IR (FIL, HOMA-IR), (b) lipid profile (TG, HDL-C, LDL-C, APO-A1, B and C3, and APO-B/APO-A1 ratio); (c) glycemic control (FG, HbA1c); (d) systemic inflammation (hs-CRP) and ferritin; (e) blood pressure; (f) body weight, and (g) adiponectin levels. NT-proBNP was also measured. All analyses were performed at the BARC Global Central Laboratory, in Ghent, Belgium. Histological grading of steatosis was evaluated according to the SAF and NASH CRN scoring systems. Steatosis was also assessed with ultrasound-based imaging (CAP™ on the FibroScan® device) using Youden cutoff values for S ≥ S1, S ≥ S2, and S ≥ S3 of 302 dB/m, 331 dB/m, and 337 dB/m, respectively[43]. All analyses were done according to the IRB-approved study documents.

### Statistical analyses
This manuscript reports the results of several secondary objectives of the NATIVE trial, that were prespecified in the protocol and the Statistical Analysis Plan (SAP), including the effect of lanifibranor 1200 mg/day and 800 mg/day versus placebo on CMH markers, evaluated after 24 weeks of treatment (EOT) compared to baseline, and for some markers also at TW4. Further post-hoc analyses, i.e., not prespecified in the SAP, included: the effects of therapy in the subgroups of patients with and without overt T2D; in patients with prediabetes at baseline, (defined as FG level between 5.6 and 6.9 mmol/L [100 to 125 mg/dL])[68]; the effects of therapy on hs-CRP categorized as high CV risk (>3 mg/L), intermediate (1–3 mg/L) or low risk (<1 mg/L)[69]; the effects of therapy on NT-proBNP, the correlation between weight changes (both continuous and by weight-change group, i.e., <2.5%, 2.5–5% and ≥5% increase from baseline) under lanifibranor and placebo, separately, and improvement in CMH markers (lipid and glucose metabolism, IR, inflammation, liver tests, DBP, hepatic steatosis grading and CAP™ imaging); the correlation between baseline levels of adiponectin (categorized as low, medium and high, defined as <5, 5–10 and >10 μg/mL, respectively, as previously described[70]) and disease severity; the correlation between changes in adiponectin levels both continuous or by category (unchanged, moderate or high defined as <1.5-fold, 1.5 to 4-fold and >4-fold change, respectively) during treatment and improvement in CMH markers; the correlation between HbA1c levels (continuous and by level of glycemic control: ≤6% [42 mmol/mol]; >6%- ≤ 7%; >7% [53 mmol/mol]) during treatment and histological MASH activity and fibrosis stage at baseline.

Descriptive statistics were presented using the mean and SD, median, interquartile ranges and minimum-maximum for continuous variables, and frequency counts and percentages for categorical variables. For statistical tests, the type-I error risk was set at 5% (2-sided) as per protocol. No multiplicity adjustments were performed for the analyses presented in this paper, that are considered exploratory.

Point estimates, 95%CIs and p-values are provided. The CIs have not been adjusted for multiple comparisons. Comparisons of continuous parameters between treatment groups at EOT were done using Mixed Model for Repeated Measures (MMRM), using the change from baseline as endpoint, the time (Weeks 4, 14 and 24), treatment, the diabetic status, the interaction (treatment*time) and the baseline value as fixed effects, a time repeated measure within each subject and an unstructured variance covariance matrix, or using Student or Wilcoxon tests depending on tests of normality. Comparison of categorical parameters between groups was done using Cochran-Mantel-Haenszel test stratified on the diabetic status at baseline, or from Chi² or Fisher tests. The correlation between continuous parameters was estimated using Spearman correlation (Rs and *p* value). SAS® software version 9.4 was used to perform all analyses.

### Reporting summary

Further information on research design is available in the Nature Portfolio Reporting Summary linked to this article.

## Data availability

All of the data necessary to reproduce the findings described here can be found in the manuscript, figures, supplementary information and the source data file, or from the corresponding author on request. Proposals should be directed to Michael.Cooreman@inventivapharma.com. All requests for data will be reviewed by the corresponding author who will make sure that they are available and consistent with participant privacy and informed consent. A response will be provided within three months. Data requestors will need to sign a data access agreement. The data will be available from six months after publication of the manuscript for a duration of a minimum of five years. Source data are provided with this paper and its supplementary information files. Source data are provided with this paper.

## Code availability

The SAS® codes that support the findings of this study are available from the corresponding author upon request. The basic SAS® codes used to edit the statistical results are provided in the supplementary information file.

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

## Acknowledgements

The study was supported and funded by Inventiva. Inventiva clinical research staff was involved in the study design, and collection and analysis of data, prepared and reviewed the manuscript for medical and scientific accuracy. The authors retained responsibility for the final decision to submit the paper for publication. Medical writing support was provided by D.R.B. (Translational Medicine Academy, Basel, Switzerland) and funded by Inventiva. We thank the patients, study investigators and their staff who participated in the clinical study described here.

## Author contributions

Conceptualization of the analyses, interpretation of results, drafting of the manuscript (M.P.C.); statistical analyses and design of figures and tables (L.D. and P.H.M.), drafting of results section and quality control (D.R.B.), management and oversight of the trial (M.B., M.F.A., S.M.F.); contribution to interpretation of data (J.B., R.P.G., F.Z., J.L.J., P.B., M.F.A., S.M.F.); critical review of the manuscript (all authors).

## Competing interests

M.P.C., L.D., P.H.M., M.B. and P.B. are employees of Inventiva. J.L.J. is a consultant for Inventiva. D.R.B. has received honoraria from Inventiva via Translational Medicine Academy. J.B. has served as a consultant for Abbott, American Regent, Amgen, Applied Therapeutic, AskBio, Astellas, AstraZeneca, Bayer, Boehringer Ingelheim, Boston Scientific, Bristol Myers Squibb, Cardiac Dimension, Cardiocell, Cardior, Cardiorem, CSL Bearing, CVRx, Cytokinetics, Daxor, Edwards, Element Science, Faraday, Foundry, G3P, Innolife, Impulse Dynamics, Imbria, Inventiva, Ionis, Lexicon, Lilly, LivaNova, Janssen, Medtronics, Merck, Occlutech, Owkin, Novartis, Novo Nordisk, Pfizer, Pharmacosmos, Pharmain, Prolaio, Regeneron, Renibus, Roche, Salamandra, Sanofi, SC Pharma, Secretome, Sequana, SQ Innovation, Tenex, Tricog, Ultromics, Vifor, and Zoll. RPG received research support or honoraria form Amgen, Anthos Therapeutics, Daiichi Sankyo, Ionis, Dr. Reddy's Laboratories, Medical Education Resources, Menarini, SAJA Pharmaceuticals, Servier, SUM-MEET, and consults for Artivion, Beckman Coulter, Daiichi Sankyo, Gilead, Inari, Inventiva, PhaseBio Pharmaceuticals, Samsung and Aventis. FZ reports personal fees from 89Bio, Abbott, Acceleron, Applied Therapeutics, Bayer, Betagenon, Boehringer, BMS, CVRx, Cambrian, Cardior, Cereno pharmaceutical, Cellprothera, CEVA, Inventiva, KBP, Merck, NovoNordisk, Owkin, Otsuka, Roche Diagnostics, Northsea, USa2, having stock options at G3Pharmaceutical and equities at Cereno, Cardiorenal, Eshmoun Clinical research and being the founder of Cardiovascular Clinical Trialists. MFA received grants from Boehringer-Ingelheim, Gilead, Hanmi, Inventiva, Madrigal, Novo Nordisk; honoraria from Clinical Care Options, Fishawack, Medscape, Terra Firma, CLDF; royalties from Up-to-Date; consults for 89Bio, BMS, Hanmi, Inventiva, Intercept, Madrigal, Merck, Novo Nordisk. SMF received grants from Astellas, Falk Pharma, Genfit, Gilead Sciences, GlympsBio, Janssens Pharmaceutica, Inventiva, Merck Sharp & Dome, Pfizer, Roche; honoraria from Abbvie, Allergan, Bayer, Eisai, Genfit, Gilead Sciences, Janssens Cilag, Intercept, Inventiva, Merck Sharp & Dome, Novo Nordisk, Promethera, Siemens; serves as consultant for Abbvie, Actelion, Aelin Therapeutics, AgomAb, Aligos Therapeutics, Allergan, Alnylam, Astellas, Astra Zeneca, Bayer, Boehringer Ingelheim, Bristoll-Meyers Squibb, CSL Behring, Coherus, Echosens, Dr. Falk Pharma, Eisai, Enyo, Galapagos, Galmed, Genetech, Genfit, Genflow Biosciences, Gilead Sciences, Intercept, Inventiva, Janssens Pharmaceutica, PRO.MED.CS Praha, Julius Clinical, Madrigal, Medimmune, Merck Sharp & Dome, Mursla Bio, NGM Bio, Novartis, Novo Nordisk, Promethera, Roche, Siemens Healthineers.
