## [Peer Review File · Nature Communications]

THE PAN-PPAR AGONIST LANIFIBRANOR IMPROVES CARDIOMETABOLIC HEALTH IN PATIENTS WITH METABOLIC DYSFUNCTION-ASSOCIATED STEATOHEPATITISREVIEWER COMMENTS

Reviewer #1 (Remarks to the Author):

The NATIVE randomized controlled trial reported significant histological improvement in addition to improvement in cardiometabolic parameters with both doses of lanifbranor.

This paper presents the results of expanded post hoc analysis of NATIVE study to further evaluate lanifbranor's effects on these cardio metabolic parameters and adds data on its effects on steatosis measured by histological semi-quantitative grading and by the controlled attenuation parameters as measured by vibration controlled elastography, diastolic blood pressure, and its effects in pre-diabetic patients and with weight changes during the study. It confirms the initial cardiometabolic benefits reported in the NATIVE study and shows these benefits persist even in the patients who gain weight during the study.

The study is clearly written and provides good quality main and supplemental figures and tables.

The main issue is the limited novelty of the findings since they confirm and expand the original published NATIVE study results

Other issues

- Please add the data on proportion and severity of weight changes in the two lanifbranor's groups to the abstract results section
- 83% of subjects were on metformin and 20% on statins, consider adjustment of the analysis not only based on diabetes, but also on metformin and statins use.

Reviewer #2 (Remarks to the Author):

In this analysis of the NATIVE trial the authors investigated the effect of lanifbranor on cardiometabolic health. They found that blood lipids, insulin, HOMA-IR, HbA1c, fasting glucose, hs-CRP, ferritin, diastolic BP and steatosis improved significantly - independent of

diabetes status - and most patients with prediabetes returned to normal FG levels with lanifibranor. Significant increases of adiponectin with lanifibranor correlated with improvement of hepatic and cardiometabolic health markers. Therapeutic effects with lanifibranor were independent of weight change.

Comments:

1. Introduction: Particularly because the authors specifically address cardiometabolic health in patients with NAFLD in this report, they should not specifically refer to 'NAFLD' being 'the hepatic manifestation of metabolic syndrome', but address that specific cardiometabolic subphenotypes of NAFLD exist, some of which are not or only weakly related to the metabolic syndrome (PMID: 34710482; PMID: 35183303). This aspect may also be relevant for the interpretation of some of the present data.
2. Where all of the analyses performed in this study pre-specified in the protocol of the NATIVE trial?
3. The authors defined prediabetes at baseline as FG level between 5.6-6.9 mmol/L (100 to 125 mg/dL). However, they should also consider subjects with an HbA1c of 5.7%-6.4% as having prediabetes.
4. The HOMA-IR should not be calculated in patients treated with sulphonylureas.
5. The authors discuss that the increase of body weight observed with lanifibranor is not detrimental for cardiometabolic health. However, considering the large focus on very effective weight-loss strategies (e.g. results from bariatric surgery in patients with NASH (PMID: 34762106, PMID: 37088093) and GLP-1 RA-based therapy (PMID: 33185364; PMID: 37622681)) in patients with NASH and obesity, it would be important knowing whether the increase of body weight with lanifibranor treatment did not result in adverse effects on cardiac function (NT-proBNP measurements can help addressing this point).
6. Most patients in the NATIVE trial were obese with a mean BMI of about 33 kg/m². As the treatment with lanifibranor may particularly be important in non-obese patients, in whom bariatric surgery and GLP-1 RA-based treatments are not the first choice, the authors should also perform their analyses specifically in the non-obese subgroup.
7. The authors should also more in detail address to what extent specifically subcutaneous fat and not visceral fat mass increased during lanifibranor treatment, in other words whether a metabolically healthy obesity, which is not associated with increased

cardiometabolic risk (PMID: 37156256) may have been induced.

8. Do the authors have measurements of waist- and hip circumferences, which may help addressing changes of body fat distribution during treatment with lanifibranor?

9. When the authors discuss the pathogenetic link between NAFLD and CVD they should also address the important field of hepatokine research, which also allows separating the contribution of hepatic steatosis from visceral obesity in the pathogenesis of cardiometabolic diseases (PMID: 36754018).

Reviewer #3 (Remarks to the Author):

This secondary analysis of data from NATIVE trial presents some interesting results on the treatment effect of lanifibranor on cardiometabolic health (table 1) but more care should be taken in the reporting of the results as some of the conclusions are not really supported by the analyses as presented especially in the supplementary information when talking about correlation between improvements and markers.

There are a few issues and clarifications to be addressed:

1. In the abstract the method should describe the methodology used for this manuscript and not the NATIVE trial. It should be made clear that this is a secondary analysis using NATIVE trial data reporting the type of analysis and variables used, and if all or a subgroup of original trial patients were included.

2. More details should be given in the statistical analysis section. More details for the mixed model with variables included as fixed and random effect, how many timepoints and what variance structure. And was the interaction treatment*time included? Also, info on software used is missing.

3. Results in the text should be reported as point estimate and 95% CI (without too much emphasis on p-value). Considering that no multiplicity adjustment was performed it should be made clear that all analyses are exploratory/hypothesis generating.

4. Figure 1. It would be useful to have a legend explaining from which analysis the p-value was obtained. And this should be done for all the tables/figures reporting p-values.

5. Line 168. It is not reported in text or figure the number of patients with high baseline TG.

6. Lines 183-186. The percentages are based on small numbers. For clarity also 95% CI should be reported (and it would show that they overlap).
7. Supplementary figure 2. Why absolute and not relative change (with exception of TG) was used?
8. Weight change categories seem a bit arbitrary (and different categories were used for treatment and placebo). I think it would be better to use weight as continuous variable to analyse weight change in relation to CMH and hepatic markers. And the conclusions in this section (like "...improved independent of weight change") are not supported by the data as only descriptive statistics by group are reported.
9. Also for adiponectin, how were cut off for categories (unchanged, moderate, high) chosen? And why was adiponectin not used as continuous variable (or was not reported) to look at the associations with the other CMH parameter and histological scores?
10. Line 219. "Lanifibranor therapy resulted in a dose-dependent adiponectin increase". Cannot really say this based on the descriptive data reported (87% and 95% patients with moderate or high increase).
11. Supplementary figure 6 does not make much sense. Is it just a comparison of HOMA-IR change at week 24 from baseline between groups? It looks like there is no significant difference.
12. Suppl. table 8 and 10. A table with only p-values is not very informative. What is the magnitude of the correlation (and CI)? And with "correlation between decrease and improvement" do you mean that you looked at the correlation between the changes in the two variables? I think it would be more appropriate to use a regression model (mixed model) approach. Line 240 "HOMA-IR decrease with lanifibranor treatment correlated with improvement" is not really supported by data presented.

REVIEWER COMMENTS

Reviewer #1 (Remarks to the Author):

The NATIVE randomized controlled trial reported significant histological improvement in addition to improvement in cardiometabolic parameters with both doses of lanifbranor.

This paper presents the results of expanded post hoc analysis of NATIVE study to further evaluate lanifbranor's effects on these cardio metabolic parameters and adds data on its effects on steatosis measured by histological semi-quantitative grading and by the controlled attenuation parameters as measured by vibration controlled elastography, diastolic blood pressure, and its effects in pre-diabetic patients and with weight changes during the study. It confirms the initial cardiometabolic benefits reported in the NATIVE study and shows these benefits persist even in the patients who gain weight during the study.

The study is clearly written and provides good quality main and supplemental figures and tables.

The main issue is the limited novelty of the findings since they confirm and expand the original published NATIVE study results

While the paper published in NEJM in October 2021 does contain a tabular listing of cardiometabolic markers at baseline and at the end of therapy, the analyses and discussion of the latter paper focus on the effects of lanifbranor on improvement of liver histological disease activity and fibrosis. The current manuscript is based on in depth analyses of the broad spectrum of cardiometabolic and hepatic health markers that span the disease biology from upstream insulin resistance to downstream hepatic fibrosis and also providing data that were not reported in the NEJM paper (such as hepatic steatosis and ferritin as a marker of systemic inflammation). Additional analyses such as the interactions between adiponectin, insulin sensitivity and cardiometabolic and hepatic health in response to therapy with a pharmacological agonist for all three PPAR isoforms also illustrates the systemic metabolic-immune nature of the disease and the relevance of therapeutic effects on all nodes of the disease beyond what was reported in the NEJM paper. The critical difference between weight gain with placebo and weight gain with lanifbranor therapy, with regard to both cardiometabolic and hepatic health, and the metabolic efficacy of lanifbranor in patients who have prediabetes are examples of the analyses in this manuscript that also provide new insights. Since the NEJM publication, the nomenclature has changed and NAFLD is now Metabolic Dysfunction Associated Steatotic Liver Disease, with a diagnosis based on the presence of cardiometabolic risk factors, further stressing the need to incorporate cardiometabolic health in the management of these patients. Hence, the data incorporated in the current paper emphasise the medical relevance of a therapeutic approach that has the potential to improve cardiometabolic as well as hepatic manifestations of MASH and are thus aligned with current efforts of professional organisations to implement the definition of NASH to Metabolic Dysfunction Associated Steatohepatitis¹. We have now also adopted the new nomenclature in the manuscript.

Other issues

- Please add the data on proportion and severity of weight changes in the two lanifbranor's groups to the abstract results section

The abstract has been updated accordingly, and results were provided for pooled lanifibranor data, due to word count limits.

- 83% of subjects were on metformin and 20% on statins, consider adjustment of the analysis not only based on diabetes, but also on metformin and statins use.

We have rerun the results taking into consideration concomitant use of metformin and statins as covariates in the MMRM models. The results are provided in the Table below; they are similar to the results shared in Table 1 of the manuscript. We have added a sentence in the results section and footnote in Table 1 mentioning that 'Similar results were obtained when considering concomitant use of metformin and statins as covariates in the statistical model (results not shown)'.

Table: Lanifibranor treatment effect versus placebo on markers of cardiometabolic health

Category	Parameters (unit)	Adjusted Mean Difference vs Placebo at week 24 (SE) 95%CI p-value	
		Lanifibranor 800 mg/day N = 83	Lanifibranor 1200 mg/day N = 83
Insulin resistance	FIL (pmol/L)	-84.27 (15.96) [-115.77; -52.78] <.0001	-80.36 (16.12) [-112.17; -48.56] <.0001
	HOMA-IR*	-4.13 (0.83) [-5.76; -2.50] <.0001	-4.18 (0.83) [-5.81; -2.54] <.0001
Glycemic control	FG (mmol/L)	-1.03 (0.17) [-1.36; -0.70] <.0001	-0.85 (0.17) [-1.17; -0.52] <.0001
	HbA1c (%)	-0.46 (0.07) [-0.60; -0.32] <.0001	-0.49 (0.07) [-0.63; -0.35] <.0001
Lipid metabolism and apolipoprotein levels	Triglycerides (mmol/L)	-0.55 (0.13) [-0.80; -0.30] <.0001	-0.50 (0.13) [-0.75; -0.25] 0.0001
	HDL-C (mmol/L)	0.16 (0.03) [0.09; 0.22] <.0001	0.10 (0.03) [0.03; 0.17] 0.0040
	LDL-C (mmol/L)	0.01 (0.10) [-0.18; 0.20] 0.9306	0.03 (0.10) [-0.16; 0.23] 0.7399
	APO-A1 (mg/dL)	-0.16 (3.15) [-6.38; 6.05] 0.9584	-4.43 (3.12) [-10.58; 1.71] 0.1565
	APO-B (mg/dL)	-10.02 (2.89) [-15.72; -4.32] 0.0007	-9.27 (2.88) [-14.94; -3.60] 0.0015
	APO-B/ APO-A1	-0.08 (0.03) [-0.13; -0.03] 0.0018	-0.06 (0.03) [-0.11; -0.01] 0.0215

Category	Parameters (unit)	Adjusted Mean Difference vs Placebo at week 24 (SE) 95%CI p-value	
		Lanifibranor 800 mg/day N = 83	Lanifibranor 1200 mg/day N = 83
	APO-C3 (ug/mL)	-18.65 (5.72) [-29.94; -7.37] 0.0013	-20.33 (5.68) [-31.53; -9.13] 0.0004
Systemic inflammation	hs-CRP (mg/L)	-1.98 (0.62) [-3.20; -0.77] 0.0015	-1.83 (0.62) [-3.05; -0.61] 0.0035
	Ferritin (µg/L)	-86.00 (21.39) [-128.18; -43.82] <.0001	-72.74 (21.41) [-114.96; -30.53] 0.0008
Liver tests	ALT (IU/L)	-24.70 (5.60) [-35.73; -13.66] <.0001	-22.53 (5.60) [-33.57; -11.50] <.0001
	AST (U/L)	-15.37 (4.66) [-24.56; -6.18] 0.0012	-11.80 (4.64) [-20.95; -2.64] 0.0119
	GGT (U/L)	-48.63 (8.20) [-64.83; -32.42] <.0001	-32.33 (8.19) [-48.51; -16.14] 0.0001
Blood Pressure (BP)	Diastolic BP (mmHg)	-4.01 (1.57) [-7.11; -0.90] 0.0117	-2.56 (1.57) [-5.65; 0.53] 0.1035
	Systolic BP (mmHg)	-2.43 (2.22) [-6.80; 1.94] 0.2748	-0.43 (2.20) [-4.78; 3.92] 0.8455
NT-pro-BNP (pmol/L)		4.18 (1.66) [0.90; 7.46] 0.0128	8.61 (1.62) [5.41; 11.81] <.0001
Steatosis	CAP™ (dB.m ⁻¹)	-16.26 (8.66) [-33.38; 0.86] 0.0626	-23.40 (9.13) [-41.46; -5.34] 0.0115

* For HOMA-IR related analyses, patients treated with sulphonylureas were removed from the analyses. Resulting from Mixed Model for Repeated Measures models using change from baseline as endpoint, the time, treatment, the diabetic status, the interaction (treatment*time) and the baseline value as fixed effects, a time repeated measure within each subject and an unstructured variance covariance matrix. Similar results were obtained when considering concomitant use of metformin and statins as covariates in the statistical model.

APO=apolipoprotein, ALT=alanine aminotransferase, AST=aspartate aminotransferase, CAP™=controlled attenuation parameter, CI=confidence interval, FG=fasting glucose, FIL=fasting insulin levels, GGT=gamma-glutamyl transferase, HDL=high density lipoprotein, HOMA=Homeostasis Model Assessment, hs-CRP=high-sensitivity C-reactive protein, LDL=low density lipoprotein, SE=standard error

Reviewer #2 (Remarks to the Author):

In this analysis of the NATIVE trial the authors investigated the effect of lanifibranor on cardiometabolic health. They found that blood lipids, insulin, HOMA-IR, HbA1c, fasting glucose, hs-CRP, ferritin, diastolic BP and steatosis improved significantly - independent of diabetes status - and most patients with prediabetes returned to normal FG levels with lanifibranor. Significant increases of adiponectin with lanifibranor correlated with improvement of hepatic and cardiometabolic health markers. Therapeutic effects with lanifibranor were independent of weight change.

Comments:

1. Introduction: Particularly because the authors specifically address cardiometabolic health in patients with NAFLD in this report, they should not specifically refer to 'NAFLD' being 'the hepatic manifestation of metabolic syndrome', but address that specific cardiometabolic subphenotypes of NAFLD exist, some of which are not or only weakly related to the metabolic syndrome (PMID: 34710482; PMID: 35183303). This aspect may also be relevant for the interpretation of some of the present data.

The reviewer refers to recent data showing that both metabolic and genetic risk factors contribute in different ways to the manifestation of NAFLD. Since the current manuscript is a post-hoc analysis of the NATIVE trial, the scope is determined by the set of data available as no samples are available for additional analyses.

With regard to genetic risk factors for NASH, we only analysed the data for the occurrence of PLPLA3 polymorphism I148M, a variant associated with more progressive NASH, including cardiometabolic manifestations. We found that the effects of lanifibranor therapy were similar for both wild type and the I148M variant; these data were recently presented and published as a poster². Since PNPLA3 polymorphisms contribute to both the metabolic (adipocyte lipase activity) and the genetic components, attributed to impaired mitochondrial function, we estimate that the results are of interest in the context of the reviewer's comment.

Inclusion criteria for the NATIVE trial required the presence of histological NASH with a certain activity score, the sum of inflammation and ballooning (cellular injury) grading; therefore, patients with metabolically healthy NAFL, such as has been described for liver steatosis associated with genetic polymorphisms without insulin resistance, would not have been eligible for the study.

With regard to the metabolic pathways that lead to NASH specifically, a large body of data documents the critical role of insulin resistance in fueling downstream pathways of altered lipid and glucose metabolism as well as inflammation. In the NATIVE trial population, 90% of enrolled patients had elevated HOMA-IR (>3).

Lastly, In order to develop NASH, in contrast to only NAFL, metabolic dysfunction is considered a prerequisite for the diagnosis according to the new nomenclature of MASH (Metabolic Dysfunction Associated Steatohepatitis). Therefore, MASH excludes patients with liver steatosis in the absence of metabolic dysfunction. In Native, 100% of patients had one or more metabolic syndrome abnormalities identified (and hence match the updated definition)

In summary, the NATIVE dataset includes patients who have NASH/MASH associated with metabolic dysfunction and does not allow the in-depth study of the NAFLD phenotype with predominantly genetic risk factors and no or little metabolic disease drivers.

2. Where all of the analyses performed in this study pre-specified in the protocol of the NATIVE trial?

We included both analyses that were pre-specified as part of secondary or exploratory objectives in the Statistical Analysis Plan and also analyses that were post-hoc. This has been specified in the section 'Methods'.

3. The authors defined prediabetes at baseline as FG level between 5.6-6.9 mmol/L (100 to 125 mg/dL). However, they should also consider subjects with an HbA1c of 5.7%-6.4% as having prediabetes.

For the definition of prediabetes we indeed opted for the widely applied definition based on fasting glucose levels as it is not influenced by factors that might influence the HbA1c accuracy, but of course acknowledge the other definition. We have done the analysis for patients with prediabetes defined by HbA1c levels as advised by the reviewer. Among the 41 patients with prediabetes at baseline (fasting glucose level between 5.6-6.9 mmol/L (100 to 125 mg/dL)), most of them had an HbA1c of 5.7%-6.4% (28 patients, 68%), 11 patients (27%) had an HbA1c<5.7% and 2 (5%) had an Hba1c >6.4% (See Table below). A total of 26 patients were classified as having normoglycemia according to the fasting glucose criteria but were classified as having prediabetes according to the HbA1c criteria (See Table below): 17 received lanifibranor and 9 were on placebo.

		Fasting Glucose		
		<5.6 mmol/L Defined as Normoglycaemics	[5.6-6.9] mmol/L Patients with pre-diabetes	Total non-T2DM patients
HbA1c	<5.7 %	57	11	68
	[5.7-6.4] %	26	28	54
	>6.4 %	0	2	2
	Total	83	41	124 non-T2DM patients

Among these 26 patients, the figure below shows that HbA1c was improved at EOT in both lanifibranor arms but remained stable under placebo, and that respectively 83% and 64% under lanifibranor 800 and 1200 mg decreased below the threshold of 5.7% versus 11% under placebo.

		800mg N=6	1200mg N=11	Placebo N=9
HbA1c status at EOT	N	6	11	9
	<5.7 pct	5 (83%)	7 (64%)	1 (11%)
	[5.7-6.4] pct	1 (17%)	4 (36%)	8 (89%)

4. The HOMA-IR should not be calculated in patients treated with sulphonylureas.

A total of 22% of patients were treated with sulphonylureas concomitantly during the study. As per recommendation of the editor, all HOMA-IR related analyses were reviewed and results were updated in the paper by removing these 22% of patients.

5. The authors discuss that the increase of body weight observed with lanifibranor is not detrimental for cardiometabolic health. However, considering the large focus on very effective weight-loss strategies (e.g. results from bariatric surgery in patients with NASH (PMID: 34762106, PMID: 37088093) and GLP-1 RA-based therapy (PMID: 33185364; PMID: 37622681)) in patients with NASH and obesity, it would be important knowing whether the increase of body weight with lanifibranor treatment did not result in adverse effects on cardiac function (NT-proBNP measurements can help addressing this point).

Patients with heart failure (Class C and D according to the American Heart Association) were excluded from participation in NATIVE. During the study, one patient who received Lanifibranor 1200 mg/d was reported by the investigator to have nontreatment-related mild heart failure that did not lead to further investigation or hospitalization; transient peripheral oedema related to study drug by the evaluation of the investigator was described in four patients, two at 800 and two at 1200 mg/d. These findings have been described in the NEJM paper referred to in the manuscript. No signs of adverse effects on cardiac function were otherwise reported.

NT-proBNP values show a slight increase in patient treated with lanifibranor and these results have been added to the manuscript (Table 1 and Figure 1J). But an increase in a biomarker level does not in itself

signify an adverse clinical effect. NT-proBNP values remain overall low and well below values that are considered diagnostic for heart failure, above 900 pg/ml.

The NT-proBNP values in lanifibranor-treated patients were at baseline between 5 pg/mL and below the upper limit normal (ULN) for all patients, except for 5 ones that had NT-pro-BNP values above the ULN at baseline which remained above the ULN at Week 24 (ranges 158 to 705 pg/mL).

NT-proBNP values in lanifibranor-treated patients at EOT were between 5 and 800 pg/mL (corresponding to the patient with 705 pg/mL at baseline).

It should be noted that reference values for normal range increase with age; within NATIVE, 19% of patients were over the age of 65 years, a patient population in whom heart failure would be associated with higher NT-pro-BNP values.

While most weight gain with PPAR γ agonist therapy has been ascribed to maturation of insulin sensitive adipose tissue, fluid retention has also been described with this mechanism of action, as discussed in the manuscript. Since only a small percentage of patient developed oedema, which was transient and did not recur after a short treatment interruption, and in the absence of haemodilution, any potential effect of Lanifibranor on fluid retention would be mild and not expected to have clinical consequence in patients with normal heart function. On the other hand, the mechanical effect may suffice to induce release of NT-proBNP from cardiomyocytes.

Overall, PPAR γ agonist therapy with pioglitazone has shown to provide beneficial therapeutic effects on cardiovascular outcomes; the cardiometabolic benefits outweigh any potential adverse effects resulting from fluid retention³. Thus, ultimately, clinical studies evaluating the effects of Lanifibranor after longer treatment duration will be needed to clarify the significance of these effect.

6. Most patients in the NATIVE trial were obese with a mean BMI of about 33 kg/m². As the treatment with lanifibranor may particularly be important in non-obese patients, in whom bariatric surgery and GLP-1 RA-based treatments are not the first choice, the authors should also perform their analyses specifically in the non-obese subgroup.

We have added the effect of treatment with lanifibranor versus placebo on markers of cardiometabolic health per BMI at baseline (obese versus non-obese). Results are consistent with the overall analyses. This has been mentioned in the results section and the data are reported in Supplementary Table 12 of the manuscript.

7. The authors should also more in detail address to what extent specifically subcutaneous fat and not visceral fat mass increased during lanifibranor treatment, in other words whether a metabolically healthy obesity, which is not associated with increased cardiometabolic risk (PMID: 37156256) may have been induced.

Subcutaneous and visceral adipose tissue were not quantified in NATIVE and hence currently we do not have the data to accurately address this very relevant question. We will, however, obtain such data from a separate study (NCT05232071), which is ongoing and includes MRI-based imaging for this purpose. The evidence for improved cardiometabolic health is provided by the improvement in insulin sensitivity and of the broad panel of pertinent markers that are the scope of this manuscript, which imply a beneficial

effect on adipose tissue, i.e. maturation toward insulin-sensitive adipose tissue. Of note, a separate exploratory study sponsored by the University of Florida demonstrated improved insulin sensitivity in liver, skeletal muscle and adipose tissue, measured by the euglycemic clamp method and the use of deuterated glucose, in patients who were treated with lanifibranor versus placebo. These data have recently been presented⁴.

8. Do the authors have measurements of waist- and hip circumferences, which may help addressing changes of body fat distribution during treatment with lanifibranor?

Body fat distribution is being measured by MRI imaging in the above-mentioned ongoing trial (NCT05232071) but, as mentioned in the reply to the previous comment, the data from NATIVE do not provide information on body fat distribution, apart from the reduction of hepatic steatosis; together with the improvement of insulin sensitivity and cardiometabolic health markers it documents improvement in metabolic health that includes adipose tissue. This is further substantiated by the robust increase in levels of adiponectin, an endogenous insulin sensitizer, in patients treated with Lanifibranor.

Waist circumference (WC) was measured at each visit by investigators in our study. No reduction WC was observed in this study (data shown below). However, as lanifibranor induces some degree of weight gain and as WC correlates with BMI⁵, we do not anticipate that the combined effects of lanifibranor on overall weight and on adipose tissue distribution imply WC would be a good indicator of body composition change with this particular mode of action. As discussed in the manuscript, a reduction in hepatic steatosis sets in with Lanifibranor therapy. Data on the distribution of visceral versus subcutaneous fat will be obtained in the ongoing trial referred to above.

Table 1. Lanifibranor treatment effect *versus* placebo on waist circumference

		Adjusted Mean Difference versus Placebo at week 24 (SE) 95%CI p-value	
Category	Parameters (unit)	Lanifibranor 800 mg/day N = 83 N _{T2D} = 33 / N _{non-T2D} = 50	Lanifibranor 1200 mg/day N = 83 N _{T2D} = 35 / N _{non-T2D} = 48
Waist circumference (cm)	Overall	2.64 (0.82) [1.03; 4.25] 0.001	1.39 (0.81) [-0.21; 2.99] 0.089
	in T2D patients	4.27 (1.33) [1.63; 6.91] 0.002	3.72 (1.30) [1.14; 6.30] 0.005

9. When the authors discuss the pathogenetic link between NAFLD and CVD they should also address the important field of hepatokine research, which also allows separating the contribution of hepatic steatosis from visceral obesity in the pathogenesis of cardiometabolic diseases (PMID: 36754018).

We appreciate the comments on the role of hepatokines, including fetuin A, FGF21, and others in the pathophysiology of steatotic liver disease, steatohepatitis, and their potential for further understanding the interaction between (visceral) adipose tissue, the liver and other endocrine organs. This is a large

and expanding emerging field. For the analysis of the NATIVE trial data, we are limited to the disease markers that have been analysed from biosamples taken within this trial. We estimate that currently ongoing and future clinical trials with lanifibranor in this field will provide an opportunity to obtain more elaborate data on hepatokines. We have addressed this relevant topic in the discussion of the manuscript.

Reviewer #3 (Remarks to the Author):

This secondary analysis of data from NATIVE trial presents some interesting results on the treatment effect of lanifibranor on cardiometabolic health (table 1) but more care should be taken in the reporting of the results as some of the conclusions are not really supported by the analyses as presented especially in the supplementary information when talking about correlation between improvements and markers.

There are a few issues and clarifications to be addressed:

1. In the abstract the method should describe the methodology used for this manuscript and not the NATIVE trial. It should be made clear that this is a secondary analysis using NATIVE trial data reporting the type of analysis and variables used, and if all or a subgroup of original trial patients were included.

The abstract has been modified accordingly.

2. More details should be given in the statistical analysis section. More details for the mixed model with variables included as fixed and random effect, how many timepoints and what variance structure. And was the interaction treatment*time included? Also, info on software used is missing.

Details have been added in the Statistical methods, and in footnotes of corresponding tables. Actually, MMRM used change from baseline as endpoint, the time (Weeks 4, 14 and 24), treatment, the diabetic status, the interaction (treatment*time) and the baseline value as fixed effects, a time repeated measure within each subject and an unstructured variance covariance matrix. SAS® software was used.

3. Results in the text should be reported as point estimate and 95% CI (without too much emphasis on p-value). Considering that no multiplicity adjustment was performed it should be made clear that all analyses are exploratory/hypothesis generating.

The exploratory nature of all analyses performed was added in Statistical methods, and the paper has been revised using preferably point estimates and 95%CI, rather than p-values.

4. Figure 1. It would be useful to have a legend explaining from which analysis the p-value was obtained. And this should be done for all the tables/figures reporting p-values.

Such details were added as footnotes in corresponding Tables/Figures.

5. Line 168. It is not reported in text or figure the number of patients with high baseline TG.

This information has been added.

6. Lines 183-186. The percentages are based on small numbers. For clarity also 95% CI should be reported (and it would show that they overlap).

This information has been added.

7. Supplementary figure 2. Why absolute and not relative change (with exception of TG) was used?

This has been corrected for TG for consistency purpose, only absolute changes are considered now.

8. Weight change categories seem a bit arbitrary (and different categories were used for treatment and placebo). I think it would be better to use weight as continuous variable to analyse weight change in relation to CMH and hepatic markers. And the conclusions in this section (like "...improved independent of weight change") are not supported by the data as only descriptive statistics by group are reported.

The categories for weight change were chosen for the purpose of illustrating the findings of the study: weight increase was shown to be metabolically healthy – improvement of all cardiometabolic and hepatic health markers improved similarly whether or not a patient had weight change. The weight gain thus does not in itself represent a clinical concern with regard to worsening of health, and as a result the categorization is in that regard arbitrarily chosen. For patients who received lanifibrator, we further divided the category 'weight gain > 2.5% from baseline' into two groups, since this represents roughly 50% of the patients who had received lanifibrator. Because overall a smaller number of patients who had received placebo had weight gain, as expected, a further sub-categorization of patients on placebo with weight gain did not make sense. The key finding is that weight gain occurring in patients who take lanifibrator is health-wise quintessentially different from weight gain that may occur in patients who took placebo, and therefore have weight gain most likely resulting from lifestyle factors.

The conclusion of the section has been reworded according to the recommendation of the reviewer.

Regarding your comment to use weight as continuous variable to analyse weight change in relation to CMH and hepatic markers, this has been actually done, in parallel of the categorical approach, to make sure both were consistent and lead to the same interpretation. As an example, please see below, for HOMA-IR the continuous and categorical analyses:

Continuous approach:

Categorical approach:
Lanifibranor

Placebo

We believe the presentation of the data as categorical variables, instead of continuous variables, adds to the clarity of the message based on the results. This is helpful in particular to demonstrate the effect of lanifibranor on insulin resistance and other cardiometabolic markers, including the fact that with lanifibranor this improvement occurs both in patients who have stable weight and in patient whose weight increases, in contrast to what is seen in patients who received placebo.

9. Also for adiponectin, how were cut off for categories (unchanged, moderate, high) chosen? And why was adiponectin not used as continuous variable (or was not reported) to look at the associations with the other CMH parameter and histological scores?

The categories for adiponectin levels chosen in the manuscript to reflect risk for cardiovascular disease, as previously described and referred to in the manuscript. In the absence of standardized cut-off values, which to our knowledge have not been established for adiponectin, this categorization provides adequacy to illustrate therapy-induced changes in adiponectin levels and their association with an improvement of cardiometabolic health.

Regarding your comment to use adiponectin as continuous variable to analyse adiponectin fold in relation to CMH parameters and histological scores, this has been actually done, in parallel of the categorical approach, to make sure both were consistent and lead to the same interpretation. As an example, please see below, for HDL-C the continuous and categorical analyses:

Continuous approach, at Week 24

Categorical approach, over time

For the same reason as given under comment #9, we have presented the data based on categorical variables.

10. Line 219. “Lanifibranor therapy resulted in a dose-dependent adiponectin increase”. Cannot really say this based on the descriptive data reported (87% and 95% patients with moderate or high increase).

We have added in Supplementary Table 5 the continuous adiponectin folds at EOT, to support this result. The adiponectin folds were higher in the lanifibranor 1200 group (mean=4.5) versus lanifibranor 800 group (mean=3.8). We have as well re-worded this paragraph.

11. Supplementary figure 6 does not make much sense. Is it just a comparison of HOMA-IR change at week 24 from baseline between groups? It looks like there is no significant difference.

We acknowledge the comment. We have decided to remove the figure.

12. Suppl. table 8 and 10. A table with only p-values is not very informative. What is the magnitude of the correlation (and CI)? And with “correlation between decrease and improvement” do you mean that you looked at the correlation between the changes in the two variables? I think it would be more appropriate to use a regression model (mixed model) approach. Line 240 “HOMA-IR decrease with lanifibranor treatment correlated with improvement” is not really supported by data presented.

Spearman correlation coefficients and 95% CI were added in the paper, as per your first comment. We appreciate the comment on the ‘correlation between decrease and improvement’. In our analyses, correlations between 2 continuous variables (eg “decrease and improvement”) were indeed assessed using non-parametric Spearman correlation approach, to get preliminary results on the relationship between variables. We assessed the strength and direction of the monotonic relationship between these variables, without any assumptions made (as the regression model would imply eg normality).

Line 240 in the manuscript has been reworded accordingly.

References

1. Rinella ME, Lazarus JV, Ratziu V, Francque SM, Sanyal AJ, et al. A multisociety Delphi consensus statement on new fatty liver disease nomenclature. *J Hepatol* 2023; 79(6):1542-1556.
2. Griffel LH, Francque SM, Abdelmalek MF, Huot-Marchand Ph, Dzen L, et al. Lanifibranor improves liver histology and markers of cardiometabolic health in patients with NASH independent of PNPLA3 genotype: a retrospective analysis of the NATIVE study. AASLD 2023, poster.
3. Liao HW, Saver JL, Wu YL, Chen TH, Lee M, et al. Pioglitazone and cardiovascular outcomes in patients with insulin resistance, pre-diabetes and type 2 diabetes: a systematic review and meta-analysis. *BMJ Open* 2017; 7(1):e013927.
4. Barb D, Kalavalapalli S, Leiva ED, Bril F, Huot-Marchand P. Lanifibranor reverses insulin resistance and improves glucose and lipid metabolism in patients with type 2 diabetes and metabolic dysfunction-associated steatotic liver disease (MASLD). AASLD 2023, poster.
5. Gierach M, Gierach J, Ewertowska M, Arndt A, Junik R. Correlation between body mass index and waist circumference in patients with metabolic syndrome. *ISRN Endocrinol* 2014; 514589.

REVIEWERS' COMMENTS

Reviewer #2 (Remarks to the Author):

The authors have well addressed the comments and questions and provide a very interesting and clinically important manuscript about the impact of lanifibranor treatment on cardiometabolic health in patients with NASH.

Reviewer #3 (Remarks to the Author):

I think the manuscript is much clearer now and my comments have been addressed.

I have a clarification about my comment on p-value. I didn't mean p-values had to be removed completely, I would report them alongside the 95% CI. And for example, in line 284: "and "ballooning improvers" (0.45%) versus "non improvers" (0.33%)" where the difference is not significant ($p=0.088$), p-value should be reported.

One minor further comment:

Suppl. Figure Fig. 2A to F. Why for HDL, CRN-S improvement yes/no category is reported? I think that, for consistency it should be presented with CRN-S evolution.

RESPONSE TO REVIEWERS' COMMENTS

Reviewer #2 (Remarks to the Author):

The authors have well addressed the comments and questions and provide a very interesting and clinically important manuscript about the impact of lanifibranor treatment on cardiometabolic health in patients with NASH.

Many thanks for the positive feedback; this is greatly appreciated.

Reviewer #3 (Remarks to the Author):

I think the manuscript is much clearer now and my comments have been addressed.

Many thanks for the positive feedback; this is greatly appreciated.

I have a clarification about my comment on p-value. I didn't mean p-values had to be removed completely, I would report them alongside the 95% CI. And for example, in line 284: "and "ballooning improvers" (0.45%) versus "non improvers" (0.33%)" where the difference is not significant ($p=0.088$), p-value should be reported.

Thanks for the clarification. We have added p-values alongside the 95%CI.

One minor further comment:

Suppl. Figure Fig. 2A to F. Why for HDL, CRN-S improvement yes/no category is reported? I think that, for consistency it should be presented with CRN-S evolution.

Thanks for your comment. The Figure 2E has been updated accordingly.